# GC4NC: A Benchmark Framework for Graph Condensation on Node Classification with New Insights

**Shengbo Gong**[*1], **Juntong Ni**[*1], **Noveen Sachdeva**[2], **Carl Yang**[1], **Wei Jin**[1]
[1]Emory University, [2]Google Deepmind
{shengbo.gong, juntong.ni}@emory.edu    noveen@google.com
{j.carlyang, wei.jin}@emory.edu

## Abstract

Graph condensation (GC) is an emerging technique designed to learn a significantly smaller graph that retains the essential information of the original graph. This condensed graph has shown promise in accelerating graph neural networks while preserving performance comparable to those achieved with the original, larger graphs. Additionally, this technique facilitates downstream applications like neural architecture search and deepens our understanding of redundancies in large graphs. Despite the rapid development of GC methods, particularly for node classification, a unified evaluation framework is still lacking to systematically compare different GC methods or clarify key design choices for improving their effectiveness. To bridge these gaps, we introduce **GC4NC**, a comprehensive framework for evaluating diverse GC methods on node classification across multiple dimensions including performance, efficiency, privacy preservation, denoising ability, NAS effectiveness, and transferability. Our systematic evaluation offers novel insights into how condensed graphs behave and the critical design choices that drive their success. These findings pave the way for future advancements in GC methods, enhancing both performance and expanding their real-world applications. The code is available at https://github.com/Emory-Melody/GraphSlim/tree/main/benchmark

## 1 Introduction

Graphs are ubiquitous data structures describing relations of entities and have found applications in various domains such as chemistry [1, 2], bioinformatics [3, 4], neuroscience [5], epidemiology [6], and e-commerce [7, 8]. To harness the wealth of information in graphs, graph neural networks (GNN) have emerged as powerful tools for exploiting structural information to handle diverse graph-related tasks [9, 10, 11, 7, 12]. However, the proliferation of large-scale graph datasets in practical applications introduce significant computational difficulties for GNN utilization [13, 14, 15]. These large datasets complicate GNN training, as time complexity escalates with the increase of nodes and edges. Furthermore, the extensive sizes of graphs also strain GPU memory, disk storage, and communication bandwidth [15].

Inspired by dataset distillation (or dataset condensation) [16, 17, 18] in the image domain, graph condensation (GC) [14, 19, 20, 21] has been proposed to learn a significantly smaller (e.g., $1,000\times$ smaller number of nodes) graph that retains essential information of the original large graph. This condensed graph is expected to train downstream GNNs in a highly efficient manner with minimal performance degradation. As a data-centric technique, GC is considered to be orthogonal to existing model-centric efforts on GNN acceleration [22, 23], since using condensed graph datasets as input can further speed up existing models. Remarkably, GC not only excels at compressing graph data

---

[*]Equal contribution

39th Conference on Neural Information Processing Systems (NeurIPS 2025) Track on Datasets and Benchmarks.

but also shows promise for various other applications, such as federated learning [24] and neural architecture search (NAS) [25].

Despite the rapid advancements in this field, the lack of a unified and comprehensive evaluation protocol for GC significantly hinders progress in evaluating, understanding and improving these methods. _First_, existing GC methods adopt different approaches to select the best condensed graphs, including variations in validation models, reliance on test set results rather than validation ones, and conducting overly frequent intermediate validations, which could introduce unfairness in evaluation. _Second_, while most GC methods are evaluated primarily on performance and transferability, they often neglect critical aspects such as the effectiveness of NAS. Furthermore, intuitive benefits of GC like privacy preservation and denoising ability are frequently mentioned but remain under-explored [26, 19]. _Third_, the impact of design choices during the condensation process including the condensation objectives, how condensed graphs are initialized, whether to generate a condensed graph structure, and which graph properties to preserve, are still poorly understood. By systematically addressing these limitations, we aim to shed light on the successes and pitfalls in current GC research and guide future directions in this evolving area. Given that most GC methods are developed for node classification (NC), we will focus on this task and propose a new benchmark framework, GC4NC, with the following contributions:

- **A Fair Evaluation Protocol.** We establish a graph condensation benchmark by introducing a fair and consistent evaluation protocol that facilitates comparison across methods. This unified evaluation approach properly utilizes validation data to select the most effective condensed graphs. In addition, we provide an open-source, well-structured, and user-friendly codebase specifically designed to facilitate easy integration and evaluation of different GC approaches.
- **Comprehensive Comparison through Multiple Dimensions.** Using the fair evaluation protocol, we conduct comprehensive comparisons of various GC methods across multiple dimensions including (a) performance and scalability, (b) privacy preservation, (c) denoising ability, (d) NAS effectiveness, and (e) transferability. To our knowledge, we are the first to systematically benchmark privacy preservation and denoising ability across various GC methods.
- **In-Depth Analysis of Design Choices.** We further conduct a thorough analysis of how key design choices impact condensation performance, including data initialization, structure-free vs. structure-based methods, and graph property preservation. Our results provide valuable guidance for optimizing and exploring these critical choices in future research.
- **Novel Insights.** Through a comprehensive comparison of these methods, our experimental results provide key insights into the behavior of graph condensation such as:
  (a) Among varied condensation objectives, methods based on trajectory matching generally deliver the best condensation performance but fall short in efficiency. Furthermore, graph condensation achieves better performance than image dataset condensation at the same reduction rates, but it struggles to scale to larger reduction rates, where **reduction rate** $r$ is defined as (#nodes in synthetic set)/(#nodes in training set).
  (b) Certain GC methods can **preserve privacy** by reducing the success of membership inference attacks while still maintaining high condensation performance.
  (c) GC methods exhibit **a certain level of denoising ability** against structural noise (both adversarial and random noise), yet they are less effective against node feature noise.
  (d) Trajectory matching or inner optimization through gradient matching is essential for reliable NAS performance and enhanced transferability.
  (e) Compared to *structure-based* methods, *structure-free* methods exhibit strong condensation performance and favorable efficiency but poorer denoising ability.

Note that two concurrent works [27, 28] on GC benchmarks have emerged alongside this paper. While all studies contribute uniquely to the field of graph condensation, **GC4NC** stands out by offering deeper insights. First, it covers a wider range of GC methods for NC. Second, it pioneers the exploration of GC methods in terms of privacy preservation and denoising ability. Third, it provides a more in-depth analysis of graph property preservation to enhance the understanding of GC methods. For further details, please refer to the Appendix A.1.

## 2 Related Work

### 2.1 Graph Condensation

Graph condensation (GC) is an emerging technique designed to create a significantly smaller graph that preserves the maximum amount of information from the original graph [14, 19, 29, 30, 31, 32]. The goal is to ensure that GNNs trained on this condensed graph exhibit comparable performance to those trained on the original one. Based on certain condensation objectives, existing GC methods employ the following matching strategies to bridge the gap between condensed and real graphs:

**Gradient Matching (GM). GCond** [14] matches gradients between the original graph $\mathcal{T}$ and the condensed graph $\mathcal{S}$ via $\min_{\mathcal{S}} \mathbb{E}_{\theta_0 \sim P_{\theta_0}} \left[ \sum_{t=0}^{T-1} D\left(\nabla_\theta \mathcal{L}_\mathcal{T}, \nabla_\theta \mathcal{L}_\mathcal{S}\right) \right]$, where $D(\cdot, \cdot)$ is a distance function. This involves **inner optimization**, where the GNN is trained on $\mathcal{S}$ during matching. This nested optimization limits scalability. **DosCond** [29] improves efficiency by matching only the first epoch. **MSGC** [31] uses multiple sparse graphs to better capture structural diversity. **SGDD** [32] injects original structural information into the synthetic graph via optimal transport.

**Trajectory Matching (TM). SFGC** [33], inspired by [34], matches GNN training trajectories using offline expert parameters: $\min_{\mathcal{S}} \mathcal{L} = \|\hat{\theta}_{t+N} - \theta^*_{t+M}\|_2^2$, where $\hat{\theta}$ and $\theta^*$ are student and expert parameters, respectively. **GEOM** [30] introduces an expanding window to adapt the matching range based on node difficulty. These methods achieve strong performance but involve high cost.

**Others.** *Distribution Matching (DM)* was adapted to graphs as **GCDM** [35], which matches average embeddings across layers between original and condensed graphs. We use its structure-free variant, **GCDMX**, due to its better performance. To reduce the cost of GM, **GCSNTK** [36] replaces inner optimization with the Graph Neural Tangent Kernel (GNTK) in a Kernel Ridge Regression (KRR) framework: $\mathcal{L}_{\mathrm{KRR}} = \frac{1}{2}\|\mathbf{y}_\mathcal{T} - \mathbf{K}_{\mathcal{T}\mathcal{S}}(\mathbf{K}_{\mathcal{S}\mathcal{S}} + \epsilon \mathbf{I})^{-1}\mathbf{y}_\mathcal{S}\|^2$, where $\mathbf{K}$ is the kernel matrix and $\mathbf{y}$ are graph labels. This is known as *meta-model matching (MM)* [26]. **GDEM** [37] uses *eigenbasis matching (EM)*, a GM variant that avoids model-induced bias. **Bonsai** [38] proposes *computation trees matching* to enable efficient and reusable training across architectures.

### 2.2 Coreset Selection and Graph Coarsening

We stress the importance of considering graph reduction methods beyond GC. **First**, recent coreset [39] and coarsening methods [40] have shown strong potential in preserving GNN performance and are essential baselines for comparison. **Second**, these methods can also serve as data initialization strategies for GC (see Section 4.6). Studying GC in isolation may overlook these important connections.

**Coreset.** Coreset selection [41] finds representative samples using specific criteria. In graphs, it selects nodes or edges to form a smaller graph. We use the following baselines: **Random**, which selects nodes uniformly at random. **KCenter** [41, 42] selects nodes to minimize the maximum distance to the nearest center in embedding space. **Herding** [43] selects nodes to minimize the gap between the mean embedding and the selected set.

**Graph Coarsening.** Graph coarsening groups nodes into supernodes to retain node information. We use the following baselines: **Averaging** (MSGC [31]) averages features of training nodes per class to form supernodes. **Virtual Node Graph (VNG)** [44] uses weighted k-means to minimize forward error and computes the adjacency via optimization. **Variation Neighbors (VN)** [45, 46] merges nodes with similar neighborhoods. *Some of the above methods are not included in the main content due to page limit.*

## 3 Benchmark Design

Our benchmark design is founded on the typical workflow for Graph Condensation (GC), which we then extend to assess performance on a broader range of applications. The core workflow comprises three key stages: 1) the initialization of synthetic nodes, 2) the condensation training process, and 3) the evaluation of the resulting synthetic graph on downstream tasks. Building upon this foundation, we further evaluate the condensed graphs on advanced applications, including NAS, robustness and privacy preservation.

Table 1: Performance of graph reduction methods under three reduction rates. We report test accuracy (%) for all datasets, except for *Yelp*, where we use F1-macro (%). The best and the second-best results, excluding the whole graph training, are marked in **bold** and underlined. *Structure-free* and *structure-based* condensation methods are marked in blue and red, respectively.

| Dataset | Reduction rate (%) | Coreset | | | | | Coarsening | | Condensation | | | | | | | | Whole |
|---|---|---|---|---|---|---|---|---|---|---|---|---|---|---|---|---|---|
| | | | | | | | | | TM | | DM | | | GM | | | |
| | | Cent-D | Cent-P | Random | Herding | K-Center | Averaging | VNG | GEOM | SFGC | GCDM | GCondX | GCond | DosCond | MSGC | SGDD | |
| *Citeseer* | 0.36 | 42.86 | 37.78 | 35.37 | 43.73 | 41.43 | 69.75 | 66.14 | 67.61 | 66.27 | 70.65 | 67.79 | 70.05 | 69.41 | 60.24 | **71.87** | 72.6 |
| | 0.90 | 58.77 | 52.83 | 50.71 | 59.24 | 51.15 | 69.59 | 66.07 | 70.70 | 70.27 | 71.27 | 69.69 | 69.15 | 70.83 | **72.08** | 70.52 | |
| | 1.80 | 62.89 | 63.37 | 62.62 | 66.66 | 59.04 | 69.50 | 65.34 | **73.03** | 72.36 | 72.08 | 68.38 | 69.35 | 72.18 | 72.21 | 69.65 | |
| *Cora* | 0.50 | 57.79 | 58.44 | 35.14 | 51.68 | 44.64 | 75.94 | 70.40 | 78.14 | 75.11 | 79.21 | 79.74 | 80.17 | **80.65** | 80.54 | 80.15 | 81.5 |
| | 1.30 | 66.45 | 66.38 | 63.63 | 68.99 | 63.28 | 75.87 | 74.48 | **82.29** | 79.55 | 80.26 | 78.67 | 80.81 | 80.85 | 80.98 | 80.29 | |
| | 2.60 | 75.79 | 75.64 | 72.24 | 73.77 | 70.55 | 75.76 | 76.03 | **82.82** | 80.54 | 80.68 | 78.60 | 80.54 | 81.15 | 80.94 | 81.04 | |
| *Pubmed* | 0.02 | 56.16 | 57.28 | 49.46 | 62.91 | 62.91 | 75.60 | 75.60 | 69.64 | 67.61 | 77.62 | 72.03 | 77.36 | 58.13 | 75.25 | **78.11** | 78.6 |
| | 0.03 | 55.61 | 62.50 | 56.10 | 69.28 | 65.59 | 75.60 | 75.72 | 76.21 | 66.89 | 76.63 | 72.05 | 78.05 | 52.70 | **78.26** | 78.07 | |
| | 0.15 | 71.95 | 73.35 | 71.84 | 75.53 | 74.00 | 75.60 | 77.53 | **78.49** | 67.61 | 77.48 | 71.97 | 76.46 | 76.45 | 78.20 | 75.95 | |
| *Arxiv* | 0.05 | 32.88 | 36.48 | 50.39 | 51.49 | 50.52 | 59.62 | 54.89 | **64.91** | 64.91 | 60.04 | 59.40 | 60.49 | 55.70 | 57.66 | 58.50 | 71.4 |
| | 0.25 | 48.85 | 47.90 | 58.92 | 58.00 | 55.28 | 59.96 | 59.66 | **68.78** | 66.58 | 60.59 | 62.46 | 63.88 | 57.39 | 64.85 | 59.18 | |
| | 0.50 | 52.01 | 55.65 | 60.19 | 57.70 | 58.66 | 59.94 | 60.93 | **69.59** | 67.03 | 60.71 | 59.93 | 64.23 | 61.06 | 65.73 | 63.76 | |
| *Flickr* | 0.10 | 40.70 | 40.97 | 42.94 | 42.80 | 43.01 | 37.93 | 44.33 | **47.15** | 46.38 | 43.75 | 46.66 | 46.75 | 45.87 | 46.21 | 46.69 | 47.4 |
| | 0.50 | 42.90 | 44.06 | 44.54 | 43.86 | 43.46 | 37.76 | 43.30 | 46.71 | 46.38 | 45.05 | 46.69 | **47.01** | 45.89 | 46.77 | 46.39 | |
| | 1.00 | 42.62 | 44.51 | 44.68 | 45.12 | 43.53 | 37.66 | 43.84 | 46.13 | 46.61 | 45.88 | 46.58 | **46.99** | 45.81 | 46.12 | 46.24 | |
| *Reddit* | 0.05 | 40.00 | 45.83 | 40.13 | 46.88 | 40.24 | 88.23 | 69.96 | **90.63** | 90.18 | 87.28 | 86.56 | 85.39 | 86.56 | 87.62 | 87.37 | 94.4 |
| | 0.10 | 50.47 | 51.22 | 55.73 | 59.34 | 48.28 | 88.32 | 76.95 | **91.33** | 89.84 | 89.96 | 88.25 | 89.82 | 88.32 | 88.15 | 88.73 | |
| | 0.20 | 55.31 | 61.56 | 58.39 | 73.46 | 56.81 | 88.33 | 81.52 | **91.03** | 90.71 | 89.08 | 88.73 | 90.42 | 88.84 | 87.03 | 90.65 | |
| *Yelp* | 0.05 | 48.67 | 46.81 | 46.08 | 46.08 | 46.07 | **55.04** | 49.24 | 52.80 | 46.20 | 50.75 | 52.44 | 52.30 | 51.10 | 52.94 | 52.02 | 58.2 |
| | 0.10 | 51.03 | 46.08 | 46.28 | 52.23 | 46.22 | **53.51** | 47.33 | 47.56 | 47.96 | 52.49 | 49.70 | 53.22 | 52.54 | 50.97 | 54.13 | |
| | 0.20 | 46.08 | 46.08 | 49.31 | 47.49 | 46.85 | 54.42 | 48.63 | 49.48 | 46.70 | **55.89** | 48.77 | 51.76 | 52.19 | 51.35 | 52.86 | |

## 3.1 Evaluation Protocol

**A Unified Evaluation Approach.** Existing GC methods vary in their evaluation strategies to identify optimal condensed graphs throughout the condensation process. **First**, some approaches utilize the GNTK as the validation model, while others employ GNNs. **Second**, some select graphs based on the best test results rather than validation results. **Third**, some assess the condensed graph at every condensation epoch, whereas others opt for periodic evaluations to conserve computational resources. Thus, a unified evaluation approach is crucial for ensuring a fair comparison. We achieve this by unifying the validation model and restricting the validation frequency, as detailed in Appendix A.3.

**Multi-Dimensional Evaluation.** Many methods overlook critical evaluation dimensions such as scalability, privacy preservation, NAS performance, and transferability. Our benchmark aims to address this gap by enabling a comprehensive comparison of GC methods across these key aspects.

*(a) Performance and Scalability.* We first attempt to reproduce and measure the basic results of all graph reduction methods within our scope. In addition to evaluating the performance of GCN in node classification, we assess their efficiency and highlight the trade-off between performance and efficiency to assist users in selecting the appropriate method based on their hardware resources. Our efficiency reports include preprocessing time, running time per epoch, total running time, peak memory, GPU memory and disk memory usage. By examining the resource consumption across various dataset sizes and reduction rates, we can also illustrate the scalability of different methods. Additionally, we also examine the condensation performance across broader reduction rates. *Summary: A good GC method should achieve good performance while also ensure high efficiency.*

*(b) Privacy Preservation.* As the downstream model is trained on a synthetic graph that differs from the original, GC may preserve a certain level of privacy by obscuring sensitive information. To evaluate this capability, we assess the resilience of GC against privacy attacks. Specifically, we apply the method from [47] to measure privacy leakage across different GC techniques. This approach employs Membership Inference Attack (MIA) to assess privacy risks, where MIA accuracy reflects the probability that an adversary can correctly identify whether a node belongs to the training or test set. For a detailed explanation of why MIA is chosen over other attack methods, please refer to Appendix A.7. *Summary:* We anticipate that the condensed graph will mitigate the exposure of sensitive training information, such as membership, thereby reducing privacy risks.

*(c) Denoising ability.* Since GC preserves the essential information of the original graph, it can potentially reduce noise present in the original graph, even though it is not specifically designed for this purpose. We hypothesize that this capability may provide GC with denoising ability against

various types of noise. To study this, we inject three types of noise to the original graph before feeding it into the GC algorithms: (1) **Feature noise**, which randomly changes features for all nodes, (2) **Structural noise**, which randomly modifies edges, and (3) **Adversarial structural noise**, which learns corrupt graph structure to degrade the performance of the GNN model. Furthermore, to examine the denoising ability of GC in two settings, transductive and inductive, we apply poisoning plus evasion corruption (i.e., corrupting both the training and test graphs) on transductive datasets, and poisoning corruption (i.e., only corrupting the training graph) on inductive datasets. *Summary: We expect GC process can mitigate noise without specific denoising design.*

*(d) Neural Architecture Search (NAS).* NAS [48, 49] is one of the most promising applications of GC. It focuses on identifying the best-performing architecture from a vast pool of models but is computationally expensive, which requires the training of numerous architectures on the full dataset. Since the condensed graph is much smaller than the whole graph, GC methods are utilized to accelerate NAS [25]. In practical situations, preserving the rank of validation results between models trained on the condensed graph and the whole graph is important because we select the best architectures based on top validation results. We argue that all the graph condensation methods should be evaluated on the NAS task because it can effectively evaluate the practical value of a condensation method. *Summary: We expect a reliable correlation in validation performance between training on the condensed graph and the whole graph to be observed.*

*(e) Transferability.* The most critical aspect of evaluating GC methods is determining whether the condensed data can be effectively used to train diverse GNNs, adhering to a data-centric perspective. Usually, condensed graphs are closely tied to the backbone GNN used during the condensation process such as GCN and SGC, potentially embedding the inductive biases of that particular GNN, which might impair their performance on other GNNs. To address this concern, we aim for condensed graphs to exhibit consistent performance across different GNNs. Some previous studies [29, 31] don't include experiments evaluating transferability across GNNs. Additionally, evaluations of various methods are often performed on different datasets or reduction rates, hindering fair comparison. Thus, we assess the performance of condensed graphs on multiple widely-used GNN models with a unified evaluation setting. *Summary: A high-quality condensed graph, like a graph in the real world, should be versatile enough to train different models.*

## 3.2 Impact of design choices

Current GC methods follow similar procedural frameworks, with multiple choices available at each intermediate stage of the process. However, the effects of these internal mechanisms, such as how different configurations or choices influence the performance and effectiveness of graph condensation, remain largely underexplored. In this benchmark, we aim to go beyond just the matching strategies discussed in Section 2.1, by thoroughly investigating the following key design choices.

**Data Initialization.** As a crucial stage in the standard procedure of GC, data initialization helps accelerate convergence and enhances final results [18]. Besides, the initialization of the condensed graph can naturally integrated with coreset selection and graph coarsening methods. Previous work primarily relies on random selection for data initialization, with only a few studies employing alternative methods such as KCenter and Averaging [30, 31]. Therefore, we aim to conduct a comprehensive study on whether different data initialization can impact the performance of GC. **Structure-Free vs. Structure-Based Methods.** Another important choice is whether to synthesize the structure. Structure-based methods including GCond, DosCond, and MSGC, utilize separate multilayer perceptrons (MLP) to generate links between nodes based on the synthetic node features. Other structure-based methods adopt different strategies, e.g, SGDD employs a structure broadcasting strategy, while GDEM aligns the eigenbasis to recover the adjacency matrix. To assist future research in making this decision, we discuss it in Section 4.1 and 4.3, as this choice shows significant differences in these two aspects.

**Graph Property Preservation.** Graph data comprises features, structures, and labels, which can be characterized by various established metrics, also known as graph properties. We aim to explore what graph properties are preserved by condensed graphs and understand the reasons behind the success of current GC methods. We select the following metrics from different aspects of a graph: **Density** (structure), **Max Eigenvalue** of Laplacian matrix (spectra), Davies-Bouldin Index (**DBI**) [50] (feature) and **Homophily** [51](structure and label) . To further incorporate structural information into DBI, we developed a new metric named **DBI-AGG** (structure and feature), which calculates DBI based on node embeddings after two rounds of GCN-like aggregation.

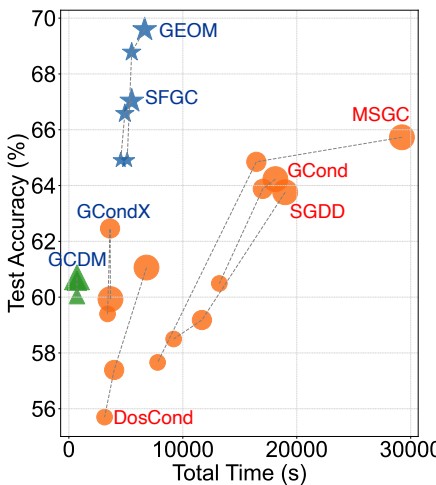

Figure 1: Test accuracy vs. total time for *structure-free* and *structure-based* condensation methods on *Arxiv*. TM is represented by ★, GM by ●, and DM by ▲. Marker sizes increase with reduction rates of 0.05%, 0.25%, and 0.50%.

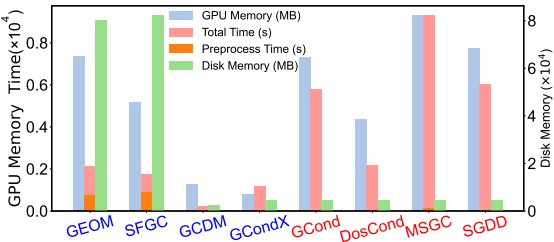

Figure 2: Comparison of GPU memory, disk memory, preprocess time, and total time on *Arxiv* ($r = 0.5\%$).

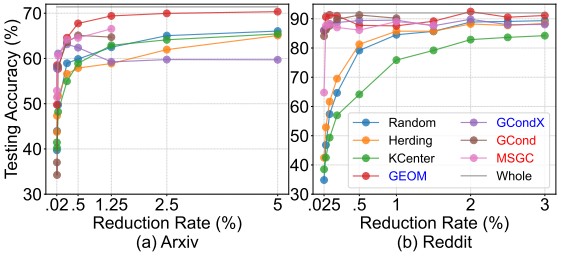

Figure 3: Varying reduction rates on Arxiv and Reddit. No mark represents OOM when the reduction rate is too large for a method.

## 4 Empirical Studies

### 4.1 Performance, Efficiency and Scalability

We provide detailed experimental setup in Appendix A.3. We report the performance of graph reduction methods in Table 1 and the efficiency in Figure 1.

**Obs. 1: TM-based methods show the best condensation performance but not the best efficiency.** From Table 1, we observe that GC methods significantly outperform coreset selection and coarsening methods and the margin is larger at low reduction rates. Among all, TM-based methods, GEOM and SFGC, lead across most datasets and reduction rates, showing the highest performance is achieved by trajectory methods. However, when we consider the efficiency and resource consumption in Figure 2, we find that though achieving state-of-the-art performance in Table 1, both GEOM and SFGC require additional preprocess time and large disk memory to produce and store the trajectory of experts. In addition, some learning-free methods, such as Averaging, exhibit high performance on certain datasets like *Yelp*, while being more efficient than all GC methods. Finally, the performance gap between the best GC methods and whole dataset training varies across datasets. Some datasets, like *Arxiv* and *Reddit*, still exhibit significant room for improvement for future GC methods.

**Obs. 2: Compared to structure-based methods, structure-free methods are more efficient while still performing well.** When comparing structure-free methods to their structure-based counterparts, such as GCondX and GCond, e.g., comparing GCondX and GCond in Figure 2 & 3 and Table 1, the following key insights emerge: (1) the absence of structure synthesis negatively impacts the performance of structure-free methods. (2) structure-based methods require significantly more memory and GPU resources, especially when applied to large graphs. (3) structure-free methods exhibit superior scalability w.r.t. reduction rates, as their computational resource usage remains relatively stable, even with increasing reduction rates. The increased complexity of structure-based methods stems from the time- and resource-intensive nature of structure synthesis, which must be repeated each time the synthetic features are updated. To fully harness the benefits of structure-based approaches, a more efficient structure generation method is needed. This is crucial as the structure provides valuable information beyond the features and has the potential to enhance the denoising ability, as discussed in Section 4.3.

**Obs. 3: GC outperforms image dataset condensation at the same reduction rate but struggles to scale effectively at larger reduction rates, where image dataset condensation excels.** We adjust

Table 2: Privacy preservation evaluation. "MIA Acc" measures how well an attacker can infer whether a node is in the training or test set. We also report node classification accuracy ("Acc"), aiming to emphasize the balance between model performance and privacy preservation.

| Methods | Cora, r = 2.6% | | Citeseer, r = 1.8% | | Arxiv, r = 0.5% | |
|---|---|---|---|---|---|---|
| | MIA Acc ($\downarrow$) | Acc ($\uparrow$) | MIA Acc ($\downarrow$) | Acc ($\uparrow$) | MIA Acc ($\downarrow$) | Acc ($\uparrow$) |
| **Whole** | 74.87 ± 1.16 | 81.50 ± 0.50 | 81.76 ± 1.01 | 72.61 ± 0.27 | 54.26 ± 0.11 | 71.43 ± 0.11 |
| **GCond** | 72.10 ± 0.96 | 80.54 ± 0.67 | 74.11 ± 0.61 | 69.35 ± 0.82 | **53.04 ± 0.18** | 64.23 ± 0.16 |
| **GCondX** | 66.83 ± 0.81 | 78.60 ± 0.31 | 71.97 ± 0.58 | 68.38 ± 0.45 | 54.64 ± 0.17 | 59.93 ± 0.54 |
| **DosCond** | 69.70 ± 0.50 | 81.15 ± 0.50 | 74.33 ± 0.34 | 72.18 ± 0.61 | 54.04 ± 0.79 | 61.06 ± 0.59 |
| **SGDD** | 70.43 ± 1.63 | 81.04 ± 0.54 | 77.07 ± 4.32 | 69.65 ± 1.68 | 53.29 ± 0.46 | 63.76 ± 0.22 |
| **GDEM** | **60.66 ± 1.26** | 81.76 ± 0.53 | 70.01 ± 2.94 | 71.74 ± 0.90 | - | - |
| **GEOM** | 67.90 ± 0.55 | **82.82 ± 0.17** | **67.55 ± 0.62** | **73.03 ± 0.31** | 53.80 ± 0.19 | **69.59 ± 0.24** |
| **SFGC** | 67.29 ± 1.02 | 80.54 ± 0.45 | 72.12 ± 0.44 | 72.36 ± 0.53 | 54.49 ± 0.53 | 67.03 ± 0.48 |

Table 3: Denoising ability evaluation. "Perf. Drop" shows the relative loss of accuracy compared to the original results before corruption. The best results are in **bold** and results that outperform whole dataset training are underlined. *Structure-free* and *Structure-based* methods are colored blue and red.

| Dataset | Method | Feature Noise | | Structural Noise | | Adversarial Structural Noise | |
|---|---|---|---|---|---|---|---|
| | | Test Acc. $\uparrow$ | Perf. Drop $\downarrow$ | Test Acc. $\uparrow$ | Perf. Drop $\downarrow$ | Test Acc. $\uparrow$ | Perf. Drop $\downarrow$ |
| *Citeseer 1.8%* | Whole | 64.07 | 11.75% | 57.63 | 20.62% | 53.90 | 25.76% |
| | GCond | **64.06** | **7.63%** | **65.64** | **5.35%** | **66.19** | **4.55%** |
| | GCondX | 61.27 | 10.40% | 60.42 | 11.65% | 60.75 | 11.15% |
| | GEOM | 58.77 | 19.53% | 51.41 | 29.60% | 57.94 | 20.67% |
| *Cora 2.6%* | Whole | 74.77 | 8.26% | 72.13 | 11.49% | 66.63 | 18.24% |
| | GCond | 67.62 | 16.04% | 63.14 | 21.61% | 68.90 | 14.45% |
| | GCondX | 67.72 | **13.85%** | 63.95 | **18.63%** | 69.24 | **11.91%** |
| | GEOM | 49.68 | 40.01% | 53.59 | 35.29% | 66.32 | 19.93% |
| *Flickr 1%* | Whole | 46.68 | 1.51% | 42.60 | 10.13% | 44.44 | 6.24% |
| | GCond | **46.29** | **1.49%** | **46.97** | **0.04%** | 43.90 | 6.58% |
| | GCondX | 45.60 | 2.11% | 46.19 | 0.83% | 42.00 | 9.83% |
| | GEOM | 45.38 | 1.63% | 45.52 | 1.32% | **44.72** | **3.06%** |

the reduction rate from values corresponding to only one node per class to values that cause OOM on large datasets and present the results in Figure 3. We use the image condensation benchmark DC-bench [18] as an analogue to our work in the graph domain. The image condensation is also trying to synthesize a much smaller number of images from original dataset while maintain the downstream task performance like image classification. While Figure 3 generally shows a positive correlation between performance and the reduction rate, we have two unique findings that are not observed in vision dataset condensation [18]: (1) GC can perform well even when condensing the graph to a single instance per class (IPC=1), whereas image condensation techniques can suffer a performance drop of 50% under similar reduction rates [18]. (2) Unlike in the image domain, GC methods cannot scale to larger IPC values due to OOM issues. We foresee the need for more scalable GC techniques, particularly those structure-based ones. In addition, our results indicate some instability of structure-free GC, as shown by $r$=0.5% on *Reddit* for GEOM and $r$=1.25% on *Arxiv* for GCondX.

## 4.2 Privacy Preservation

This attack reveals which samples were used in training, leading to privacy leakage of training set. It leverages confidence scores, i.e., the probability of the true label, to identify if a sample was part of the training set. The optimal threshold is determined by analyzing all confidence scores to maximize the attack's success in distinguishing between training and non-training samples.

**Obs. 4: Certain GC Methods such as TM and eigen-decomposition -based ones can achieve a strong balance between privacy preservation and condensation performance.** The results in Table 2 suggest the following: (1) compared to non-protected whole dataset training, GC methods enhance membership privacy by around 5%-10% on *Cora* and *Citeseer*. Notably, GDEM achieves significant preservation performance on *Cora*, with an improvement up to 14.21%, while still maintain a good performance (Acc). Also, certain method such as GEOM achieve both lowest MIA Acc and

Table 4: NAS evaluation. Best result is in **bold**. Runner-up is underlined. **Worst** is colored red.

|  | **Random** | **K-Center** | **GCondX** | **SFGC** | **GEOM** | **GCond** | **DosCond** | **MSGC** | **Whole** |
|---|---|---|---|---|---|---|---|---|---|
| Top 1 (%) | 81.88 | 81.74 | 81.49 | **82.42** | 82.19 | 81.82 | 81.91 | 82.40 | 82.51 |
| Acc. Corr. | 0.56 | 0.47 | 0.40 | **0.72** | 0.65 | 0.70 | 0.14 | 0.71 | - |
| Rank Corr. | 0.64 | 0.60 | 0.57 | 0.71 | 0.74 | 0.66 | 0.20 | **0.78** | - |

highest Acc on *Citeseer*, highlight the nature of GC in reducing the risk of privacy leakage. These improvements stem from the fact that no real training nodes are used when we apply GC, ensuring the membership information remains protected. In addition, the gain in *Arxiv* is not as significant, and we conjecture that it's close to the lower bound of 50%, resulting in a smaller margin of improvement. (2) Different reduction methods vary in their effectiveness. For example, GEOM and GDEM exhibit a strong balance between mitigating MIA accuracy and maintaining model performance. This suggests the potential to design improved GC methods that do not compromise privacy. In other words, the typical tradeoff between utility and privacy preservation could potentially be eliminated through the use of GC techniques.

### 4.3 Denoising ability

To explore the denoising ability of GC methods, specifically their ability to mitigate noise from the original graph via the condensation process, we inject three types of representative noise as outlined in Section 3.1. These include: (1) **Feature Noise**: We simulate feature noise by masking node features to zero. (2) **Structural Noise**: This is introduced by randomly adding edges to the graph. (3) **Adversarial Structural Noise**: We employ PR-BCD [52], a scalable adversarial noise using Projected Gradient Descent (PGD). In transductive settings, we apply both poisoning and evasion corruptions, which affects both the training and test phases of the graph. The perturbation rates are set to 50% for feature and structural noise and 25% for adversarial structural noise, respectively. Each corruption is repeated three times, producing three distinct corrupted graphs. We then evaluate and report the average performance across these graphs.

**Obs. 5: GC methods exhibit a certain level of denoising ability against structural noise, with structure-based approaches offering superior denoising compared to structure-free ones.** As shown in Table 3, GC methods outperform GCN trained on the whole corrupted graph in the two structural noises, but GC does not show denoising ability against feature noise. For example, GC methods achieve the highest Test Acc. across three datasets under structural noise but fall short when dealing with feature noise. This suggests that GC methods are more effective at handling structural denoising than feature denoising. Additionally, the state-of-the-art methods GEOM and the structure-free version of GCond, GCondX show lower performance compared to GCond after being corrupted, indicating that structure-free methods lose some denoising ability if they do not synthesize the structure. While GC can mitigate some noise, it still lacks specialized denoising mechanisms to achieve stronger denoising capabilities.

### 4.4 Neural Architecture Search

As a key application of GC, we evaluate the performance of NAS using three commonly-used metrics: Top 1 test accuracy, correlation between validation set accuracies, and correlation between ranks of validation set accuracies of the condensed graph and the whole graph. We use the Pearson coefficient [53] to quantify the correlation. We conduct NAS with APPNP, a flexible GNN model whose structure can vary by using a different number of propagation layers, residual coefficients, etc. More details are provided in the Appendix A.9.

**Obs. 6: Trajectory matching or inner optimization is essential for reliable NAS effectiveness.** The results in Table 4 demonstrate that: (1) GC methods demonstrate a strong potential to identify the best architectures, sometimes even outperforming the results obtained from the original dataset. (2) Methods utilizing trajectory matching demonstrate strong results in NAS. (3) Models without inner optimization during the condensation process, such as DosCond, yield poor NAS performance, with a Pearson correlation coefficient below 0.6. Given that methods employing trajectory matching or inner optimization tend to achieve better NAS results, we hypothesize that explicitly mimicking the training trajectory of GNNs is critical for effective NAS.

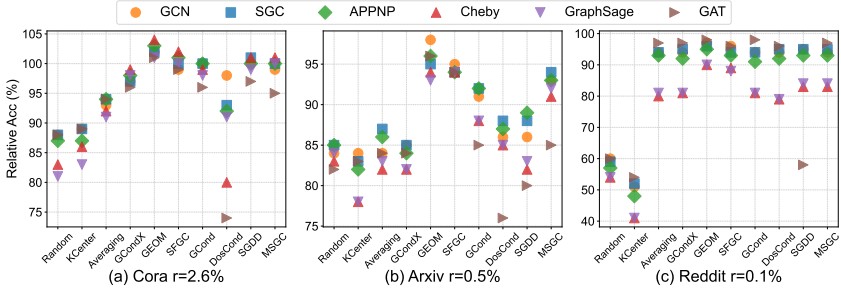

Figure 4: Condensed graph performance evaluated by different GNNs. The **relative accuracy** refers to the accuracy preserved compared to training on the whole dataset.

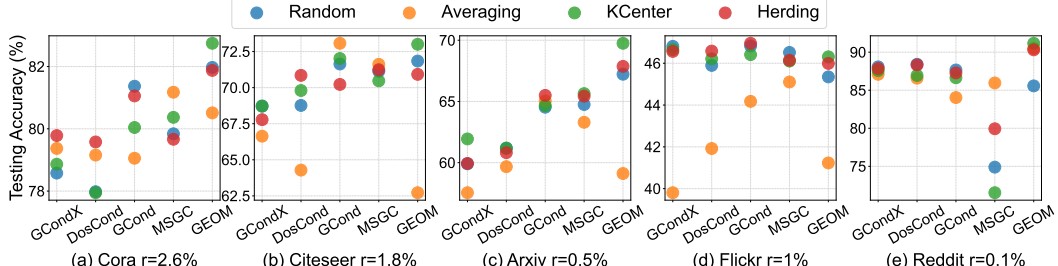

Figure 5: Test accuracy for different methods with different initialization.

## 4.5 Transferability

We conduct extensive experiments assessing the performance of condensed graphs on six widely-used GNN models: GCN [9], SGC [22], APPNP [54], Cheby [55], GraphSage [13] and GAT [10]. We tune hyperparameters for these evaluation GNN models, with the search space for hyperparameters and sensitivity analysis listed in Appendix A.8. To simplify, we fix the reduction ratios at 2.6%, 0.5%, and 0.1% for *Cora*, *Arxiv* and *Reddit*, respectively.

**Obs. 7: Different GC methods exhibit varying degrees of transferability across datasets, leaving considerable room for improvement in this area.** From Figure 4 we can observe that (1) there is no significant performance loss for the majority of cases when condensed graphs are transferred to various GNNs. This highlights the success of GC methods, which typically only use GCN or SGC for condensation. (2) However, for some methods such as DosCond and SGDD, GAT performs much worse than other GNNs. We conjecture this is because GAT is more structure-sensitive and can only leverage the connection information instead of the edge weights. (3) We also investigate the transferability to Graph Transformer [56] in Appendix A.8. However, the performance of Graph Transformer drops a lot compared to message-passing GNNs, which suggests that future research should explore the transferability to non-message-passing graph learning architectures.

**Obs. 8: Trajectory matching or inner optimization facilitates transferability.** GEOM and SFGC achieve significantly better performance than GCondX. Similarly, GCond outperforms DosCond. These two phenomena indicate that trajectory matching or inner optimization is key to improving transferability. We conjecture these two designs introduce additional inductive biases related to the backbone models used in the condensation process, which likely benefit all message-passing GNNs.

## 4.6 Data Initialization

To study the impact of different data initialization strategies, we equip 5 GC methods with 5 initialization strategies across all datasets. **Obs. 9: Current initialization strategies do not have a consistent impact across all datasets or GC methods.** Figure 5 illustrates that there is no single best data initialization method for every GC method or dataset. Notably, KCenter is the average best initialization method for most datasets. Averaging is a very unstable strategy, especially for large datasets, and it only works in rare cases. We conclude that GC methods do not need to be consistently good with different initialization strategies. Therefore, we recommend treating initialization strategies

Table 5: Graph properties of condensed graphs on Cora from different structure-based GC methods. The "**Corr.**" row shows the correlation of certain property between the condensed graph and the whole graph across five datasets. The "Whole" column of the "**Corr.**" row displays the average correlation value of the four methods.

| Graph Property | | **VNG** | **GCond** | **MSGC** | **SGDD** | **Avg.** | **Whole** |
|---|---|---|---|---|---|---|---|
| Density% | *Cora* | 52.17 | 82.28 | 22.00 | 100.00 | 64.11 | 0.14 |
| (Struc.) | Corr. | -0.81 | 0.07 | 0.55 | 0.13 | -0.02 | - |
| Max Eigenvalue | *Cora* | 3.73 | 34.90 | 1.69 | 14.09 | 13.60 | 169.01 |
| (Spectra) | Corr. | 0.85 | 0.25 | 0.95 | 0.28 | 0.58 | - |
| DBI | *Cora* | 3.69 | 1.84 | 0.70 | 4.34 | 2.64 | 9.28 |
| (Label & Feature) | Corr. | 0.81 | 0.93 | 0.94 | 0.97 | **0.91** | - |
| DBI-AGG | *Cora* | 3.59 | 0.38 | 0.57 | 0.18 | 1.18 | 4.67 |
| (Label & Feat. & Stru.) | Corr. | 0.99 | 0.93 | 0.95 | 0.89 | **0.94** | - |
| Homophily | *Cora* | 0.14 | 0.16 | 0.19 | 0.13 | 0.16 | 0.81 |
| (Label & Struc.) | Corr. | -0.83 | -0.68 | -0.46 | -0.80 | -0.69 | - |

as hyperparameters in future studies. **Obs. 10: Better coreset selection methods do not guarantee better GC initialization.** When we compare Figure 5 with coreset and coarsening columns in Table 9, we find that the best one, Herding, is not necessarily the best data initialization method for GC. This finding cautions that future research should carefully combine different graph reduction methods.

### 4.7 Graph Property Preservation

We explore the relationship between graph property preservation and structure-based GC methods. We calculate the metrics related to different graph properties for the condensed graph.

**Obs. 11: Only the properties related to node features and aggregated features, i.e., *DBI* and *DBI-AGG*, are relatively preserved in condensed graphs.** Despite examining various graph-size-agnostic graph properties, our results in Table 5 show that none of the absolute values tend to be preserved. Consequently, we resort to the *Pearson correlation* between metrics in the original and condensed graphs. From the results, we can conclude that only *DBI* and *DBI-AGG* are relatively preserved, as they have average correlation coefficients of 0.91 and 0.94. Therefore, we suggest that researchers explicitly preserve these two properties to potentially bolster performance. Notably, we observed that MSGC preserves the maximum eigenvalue up to 0.94. As further evidence, the latest method, GDEM [37], focuses on learning to preserve eigenvectors, supporting the idea that maintaining spectral properties may be beneficial. In contrast, *Density* appears to be the least important property to preserve among these GC methods. Additionally, we observe that a homophilous graph is often condensed into a heterophilous graph while still achieving high performance. This finding suggests that the relationship between GNN performance and homophily [57, 58] need to be reconsidered. **Obs. 12: Preserving Density%, DBI, and Homophily tends to be beneficial for downstream tasks.** As shown in Table 17, the *Correlation* column reports the Pearson correlation between average test accuracy and each property metric. Density%, DBI, and Homophily exhibit stronger correlations with downstream performance, while Max Eigen and DBI-AGG show weaker associations.

## 5 Conclusion and Outlook

This paper establishes the first benchmark for GC methods with multi-dimension evaluation, providing novel insights on privacy preservation, denoising ability, and design choices of current GC methods. The findings from our experimental results inspire the following future directions:

(1) **Better performance and scalability.** Future work can focus on closing the gap between GC methods and whole dataset training, and scaling to larger datasets and higher reduction rates.

(2) **Comprehensive Privacy Preservation.** Future work can exploit the privacy preservation advantage of GC methods to synthesize graphs that safeguard additional types of privacy.

(3) **Stronger Denoising Ability.** Future work can further explore the denoising ability of graph condensation methods under diverse settings, such as feature attacks and out-of-distribution (OOD) and develop techniques to enhance their robustness. Furthermore, it would also be of interest to incorporate GNN defense methods to enhance the denoising ability of GC methods.

(4) **Leveraging coreset selection or coarsening.** Future work can combine powerful coreset selection and graph coarsening methods, making GC competitive in both efficiency and performance.

## Acknowledgement

This research was supported by the U.S. National Science Foundation under Award No. 2504088. The first authors, Shengbo Gong and Juntong Ni, gratefully acknowledge Tingting Qi and Nan Li for their valuable support and encouragement during the preparation of this paper.

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

# A Appendix

## A.1 Comparison with concurrent benchmarks

To better illustrate the differences of scope and details of our benchmark and others, we create the table below:

Table 6: Comparison between our GC4NC and two concurrent works. "OOM" means if the benchmark explore when the GC methods report out-of-memory error. In "Impact of Initialization".

| Benchmark Scope | GCondenser [27] | GC-Bench [28] | GC4NC |
|---|---|---|---|
| **Methods** | | | |
| *Coreset & Sparsification* | Random, KCenter | Random, KCenter, Herding | Cent-D, Cent-P, Random, KCenter, Herding, TSpanner |
| *Coarsening* | - | - | Averaging, VNG, Clustering, VN |
| *Condensation ↓* | | | |
|   Gradient Matching | GCond, DosCond, SGDD | GCond, DosCond, SGDD | GCond, DosCond, SGDD, MSGC |
|   Trajectory Matching | SFGC | SFGC, GEOM | SFGC, GEOM |
|   Others | GCDM, DM, GDEM | GCDM, DM, KiDD, Mirage | GCDM, GDEM, GCSNTK |
| **Tasks** | Node classification | Node classification, link prediction, node clustering, graph classification | Node classification |
| **Evaluation Protocols** | | | |
| Performance on standard condensation rate | ✓ | ✓ | ✓ |
| Efficiency & Scalability | Time | Time, Memory, OOM | Time, Memory, Disk Space, OOM |
| Transferability | Cross-model | Cross-model (include GraphTransformer) | Cross-model (include GraphTransformer) |
| Privacy preservation | - | - | ✓ |
| Denoising Ability | - | - | ✓ |
| Neural Architecture Search | - | - | ✓ |
| Continual learning | ✓ | - | - |
| **Impact of inner mechanism** | | | |
| Impact of if synthesizing the structure | ✓ | ✓ | ✓ |
| Impact of Initialization | 3 strategies | 5 strategies | 5 coreset and coarsening strategies |
| Impact of validators | ✓ | - | - |
| Graph properties | - | Homophily | Density, Eigenvalue, DBI, DBI-AGG and Homophily |

From this table, our contributions are evident. **First**, we incorporate a broader range of traditional coreset and coarsening methods, along with additional condensation methods focused on node classification (NC). **Second**, we provide a more comprehensive analysis of efficiency and scalability, including disk space considerations. **Third**, we explore the application of GC methods in terms of privacy preservation and denoising effects. **Finally**, our data initialization aligns with the coreset and coarsening methods, resulting in elegant, reusable code and enabling a preliminary trial of multi-layer condensation.

Table 6 may also show some limitations of our benchmark, though most of these stem from differences in opinion and focus.

- As our title suggests, GC4NC is primarily a benchmark for NC, since the majority (approximately 90%) of condensation papers have concentrated on this task. That's why we do not include graph classification method such as KiDD [59] and we have fewer datasets compared to GC-Bench.
- We argue that the condensation model and validator can be viewed as hyperparameters, similar to how methods like GEOM approach it. Therefore, we do not study the impact of them as they are just selected by datasets.
- With regard to another important application, Continual Learning (CL), GCondenser [27] points out that many existing methods, including GDEM, SFGC, and GEOM, are incompatible with graph continual learning frameworks. This somewhat lowers the priority of CL as they are most competitive ones.

## A.2 Limitations and Future Directions

We anticipate that our benchmark and insights will contribute to progress in the field and encourage the development of more practical GC methods going forward. However, GC4NC is not without limitations and some areas of benchmarking can be further explored. These include examining the effectiveness of other privacy techniques such as Differential Privacy [60], evaluating denoising ability against other types of attacks, measuring NAS effectiveness in larger architecture spaces such as Graph Design Space [61], examining the transferability of condensed knowledge to various domains and downstream tasks, and identifying and preserving certain graph properties. Our heterophily analysis was performed on a single dataset due to the well-known scarcity of publicly available,

Table 7: Datasets Statistics

| Dataset | #Nodes | #Edges | #Classes | #Features | #Training/Validation/Test |
|---------|--------|--------|----------|-----------|----------------------------|
| *Citeseer* | 3,327 | 4,732 | 6 | 3,703 | 120/500/1000 |
| *Cora* | 2,708 | 5,429 | 7 | 1,433 | 140/500/1000 |
| *Pubmed* | 19,717 | 88,648 | 3 | 500 | 60/500/1000 |
| Arxiv | 169,343 | 1,166,243 | 40 | 128 | 90,941/29,799/48,603 |
| Flickr | 89,250 | 899,756 | 7 | 500 | 44,625/22,312/22,313 |
| Reddit | 232,965 | 57,307,946 | 210 | 602 | 15,3932/23,699/55,334 |
| Yelp | 45,954 | 3,846,979 | 2 | 32 | 36,762/4,596/4,596 |

large-scale heterophilous graphs. Additional areas for exploration based on our benchmark include assessing the impact of data augmentation at various stages of GC and examining the influence of different evaluation models.

## A.3 Experiments Setup

In an attempt to address unfairness in this area, we unify some of the settings in GC papers while leaving other hyperparameters as reported in their papers or source code. **First**, we restrict one set of hyperparameters for each dataset, ensuring that they do not vary across different reduction rates. For methods that do not follow this setting, we use the set of hyperparameters from the highest reduction rate. This setting is more practical because tuning for every reduction rate can be very expensive. **Second**, we set the evaluation interval to the number of epochs divided by 10 to balance the frequency of intermediate evaluations and total epochs for each method. This strategy will benefit fast-converging and stable methods while penalizing those that rely on long epochs and frequent validation. **Third**, we adopt GCN in all evaluation parts, training a 2-layer GCN with 256 hidden units on the reduced graph. We then evaluate it on the validation and test sets of the original graph, using 300 epochs without early stopping. We select condensed graphs with best validation accuracy for final evaluation. To mitigate the effect of randomness, we run each evaluation 10 times and report the average performance. The above GNN training settings are applied across intermediate, final evaluations, and all other experiments. **Additionally**, sparsification is only applied to the final evaluation, with the threshold adhering to the reported results in the original paper. Specifically, for structure-free methods, an identity matrix is used as the adjacency matrix during training stage. Then, in inference stage, the original graph is input into the trained model. To benchmark methods under both transductive and inductive settings, we use the former for Citeseer, *Cora* [9], *Pubmed* [62] and *Arxiv* [63], and the latter for *Flickr*, *Reddit* [64] and *Yelp* [65]. All data preprocessing and training/validation/test set splits follow the GCond paper [14]. For datasets not used in GCond paper, we follow the settings of SGDD paper [32]. More details about datasets and implementation are in Appendix A.4 and A.5.

## A.4 Datasets

We evaluate all the methods on four transductive datasets: *Cora*, *Citeseer*, *Pubmed* and *Arxiv*, and three inductive datasets: *Flickr*, *Reddit* and *Yelp*. The **reduction rate** is calculated by ( number of nodes in condensed graph) / (number of nodes in training graph). Specifically, the training graph is defined as the whole graph in transductive datasets, and only the training set for inductive datasets. Dataset statistics are shown in Table 7.

For the choices of reduction rate $r$, we divide the discussion into two parts: for transductive datasets (i.e. *Citeseer*, *Cora* and *Arxiv*), their training graph is the whole graph. For *Citeseer* and *Cora*, since their labeling rates of training graphs are very small (3.6% and 5.2%, respectively), we choose $r$ to be {10%, 25%, 50%} of the labeling rate. For *Arxiv*, the labeling rate is 53% and we choose $r$ to be {1%, 5%, 10%} of the labeling rate; for inductive datasets (i.e. *Flickr*, *Reddit* and *Yelp*), the nodes of their training graphs are all labeled (labeling rate is 100%). Thus, the fraction of labeling rate is equal to the final reduction rate $r$. The labeling rate, fraction of labeling rate and final reduction rate $r$ of each dataset are shown in Table 8.

Table 8: Explanation of Reduction Rate under transductive and inductive settings

| Dataset | Labeling Rate | Reduction Rate of Labeled Nodes | Reduction Rate $r$ |
|---|---|---|---|
| *Citeseer* | 3.6% | 10% | 0.36% |
| | | 25% | 0.9% |
| | | 50% | 1.8% |
| *Cora* | 5.2% | 10% | 0.5% |
| | | 25% | 1.3% |
| | | 50% | 2.6% |
| *Pubmed* | 0.3% | 1% | 0.3% |
| | | 10% | 3% |
| | | 50% | 15% |
| *Arxiv* | 53% | 1% | 0.05% |
| | | 5% | 0.25% |
| | | 10% | 0.5% |
| *Flickr* | 100% | 0.1% | 0.1% |
| | | 0.5% | 0.5% |
| | | 1% | 1% |
| *Reddit* | 100% | 0.05% | 0.05% |
| | | 0.1% | 0.1% |
| | | 0.2% | 0.2% |
| Yelp | 100% | 0.05% | 0.05% |
| | | 0.1% | 0.1% |
| | | 0.2% | 0.2% |

## A.5   Implementation Details

Since the node selection of Random, KCenter, and Herding varies too much in each random seed, we run these three methods three times, and all the results in Table 1 represent the average performance. We conduct all the experiments on a cluster mixed with NVIDIA A100, V100, K80 and RTX3090 GPUs. Notably, GDEM can only be reproduced by RTX3090 with their provided eigendecomposition. We use Pytorch (modified BSD license) and PyG [66] (MIT license) to reproduce all those methods in a user-friendly and unified way.

**How other variables are held constant in each comparison?**   We have used a unified pipeline (Section 4.1) where each GC method is paired with its author-recommend hyperparameters including the initialization. The resulting condensed graph is then evaluated with a standardized GCN architecture and standard GCN hyperparameters, ensuring a fair comparison of the condensed graphs themselves.

- **Privacy and NAS:** We have mentioned we use the condensed graph in Line 156 and Line 178. For most subsequent analyses, we use the fixed condensed graphs from the main evaluation. We then vary only the specific evaluation protocol (e.g., the membership inference classifier for privacy).

- **Initialization and Denoising:** For the former, we fix the other parts of condensation methods following the Section 4.1 and vary only the initialization. For the latter, we fix the entire pipeline but vary the input data by adding noise.

## A.6   Performance and Scalability

Table 9 provides the complete average accuracy with the standard deviation of 10 runs results. GDEM's results are included here but not in main content due to its reproducibility challenges on larger datasets. We also append two coreset selection baselines first introduced by [40]: **Cent-D** selects nodes based on their degree, prioritizing those with the highest connectivity. **Cent-P** [67] selects nodes with high PageRank [68] values, prioritizing those that are more central and influential in the graph structure. We also explore the potential of one traditional sparsification method called **TSpanner** [69] which only reduces the number of edges and preserves the shortest distance property.

Note that due to the reproducibility challenges of GDEM on larger datasets in our experiments, we have focused on its performance with the three small datasets and have not included it in the main content.

Table 9: Test accuracy and standard error of each graph reduction method across different datasets and three representative reduction rates for each dataset. The best and second-best results, excluding the whole graph training results, are marked in **bold** and underlined, respectively.

| Dataset | Reduction rate (%) | Coreset & Sparsification | | | | | | | Coarsen | | Condensation | | | | | | | | | | Whole |
| | | | | | | | | | | | Structure-free | | | | | Structure-based | | | | | |
| | | Cent-D | Cent-P | Random | Herding | K-Center | TSpanner | Averaging | VN | VNG | GEOM | SFGC | GCSNTK | GCDMX | GCondX | GCond | DosCond | MSGC | SGDD | GDEM | |
| Citeseer | 0.36 | 42.86±2.7 | 37.78±1.3 | 35.37±2.8 | 43.73±1.6 | 41.43±1.4 | 71.83±0.3 | 69.75±0.6 | 34.32±5.9 | 66.14±1.9 | 67.61±0.7 | 66.27±0.8 | 63.51±1.9 | 70.65±0.5 | 67.79±0.7 | 70.05±2.1 | 69.41±0.8 | 60.24±6.0 | **71.87**±0.6 | 67.88±1.8 | 72.6 |
| | 0.90 | 58.77±0.5 | 52.83±0.4 | 50.71±0.8 | 59.24±0.4 | 51.15±1.1 | 71.62±0.4 | 69.59±0.5 | 40.14±5.3 | 66.07±0.4 | 70.70±0.5 | 70.27±0.7 | 62.91±0.8 | 71.27±0.6 | 69.69±0.5 | 69.15±1.2 | 70.83±0.4 | **72.08**±0.7 | 70.52±0.6 | 70.13±1.1 | |
| | 1.80 | 62.89±0.4 | 63.37±0.4 | 62.62±0.6 | 66.66±0.5 | 59.04±0.9 | 71.60±0.4 | 69.50±0.6 | 41.98±7.0 | 65.34±1.4 | **73.03**±0.3 | 72.36±0.5 | 63.90±3.4 | 72.08±0.2 | 68.38±0.5 | 69.35±0.8 | 72.18±0.6 | 72.21±0.4 | 69.65±1.7 | 71.74±0.9 | |
| Cora | 0.50 | 57.79±1.7 | 58.44±1.7 | 35.14±2.5 | 51.68±2.1 | 44.64±4.4 | 79.79±0.4 | 75.94±0.7 | 24.62±5.7 | 70.40±0.6 | 78.14±0.5 | 75.11±2.2 | 71.58±0.9 | 79.21±0.4 | 79.74±0.5 | 80.17±0.8 | **80.65**±0.6 | 80.54±0.3 | 80.15±0.5 | 54.76±4.5 | 81.5 |
| | 1.30 | 66.45±2.2 | 66.38±1.7 | 63.63±1.3 | 68.99±0.7 | 63.28±1.4 | 80.84±0.3 | 75.87±0.6 | 51.07±5.8 | 74.48±0.6 | **82.29**±0.6 | 79.55±0.3 | 71.22±2.6 | 80.26±0.3 | 78.67±0.4 | 80.81±0.5 | 80.85±0.4 | 80.98±0.5 | 80.29±0.4 | 72.87±1.8 | |
| | 2.60 | 75.79±0.7 | 75.64±1.6 | 72.24±0.6 | 73.77±0.9 | 70.55±1.4 | 80.41±0.3 | 75.76±1.1 | 56.75±5.4 | 76.03±0.4 | **82.82**±0.2 | 80.54±0.5 | 73.34±0.6 | 80.68±0.3 | 78.60±0.3 | 80.54±0.7 | 81.15±0.5 | 80.94±0.4 | 81.04±0.5 | 81.76±0.5 | |
| Pubmed | 0.02 | 56.16±2.6 | 57.28±1.2 | 49.46±1.6 | 62.91±1.5 | **79.18**±0.2 | 62.91±1.5 | 74.09±0.6 | 75.00±0.4 | 75.60±0.4 | 69.64±1.4 | 67.61±2.0 | 29.45±10.9 | 77.62±0.2 | 72.03±1.4 | 77.36±0.7 | 58.13±2.2 | 75.25±0.7 | 78.11±0.3 | 77.52±0.7 | 78.6 |
| | 0.03 | 55.61±1.6 | 62.50±1.0 | 56.10±1.5 | 69.28±1.6 | 65.59±2.4 | **79.39**±0.3 | 75.60±0.4 | 74.09±0.6 | 75.72±0.4 | 76.21±0.7 | 66.89±3.3 | 68.37±3.0 | 76.63±1.2 | 72.05±1.6 | 78.05±0.3 | 52.70±0.3 | 78.26±0.3 | 78.07±0.3 | 78.05±1.3 | |
| | 0.15 | 71.95±0.5 | 73.35±0.4 | 71.84±0.7 | 75.53±0.4 | 74.00±0.2 | 78.39±0.2 | 75.60±0.4 | 73.68±0.5 | 77.53±0.5 | 78.49±0.2 | 67.61±4.1 | 69.89±2.2 | 77.48±0.5 | 71.97±0.5 | 76.46±0.5 | 76.45±0.4 | 78.20±0.2 | 75.95±0.3 | 78.76±1.1 | |
| Arxiv | 0.05 | 32.88±2.7 | 36.48±2.0 | 50.39±1.4 | 51.49±0.7 | 50.52±0.5 | - | 59.62±0.4 | OOM | 54.89±0.3 | **64.91**±0.6 | 64.91±0.5 | 58.21±1.7 | 60.04±0.4 | 59.40±0.5 | 60.49±0.4 | 55.70±0.3 | 57.66±0.4 | 58.50±0.2 | - | 71.4 |
| | 0.25 | 48.85±1.1 | 47.90±0.8 | 58.92±0.3 | 58.00±0.5 | 55.28±0.6 | - | 59.96±0.3 | OOM | 59.66±0.2 | **68.78**±0.6 | 66.58±0.3 | 59.98±1.7 | 60.59±0.4 | 62.46±0.3 | 63.88±0.2 | 57.39±0.2 | 64.85±0.3 | 59.18±0.2 | - | |
| | 0.50 | 52.01±0.5 | 55.65±0.5 | 60.19±0.5 | 57.70±0.2 | 58.66±0.4 | - | 59.94±0.3 | OOM | 60.93±0.2 | **69.59**±0.2 | 67.03±0.5 | 54.73±5.0 | 60.71±0.7 | 59.93±0.4 | 63.23±0.2 | 61.06±0.6 | 65.73±0.2 | 63.76±0.2 | - | |
| Flickr | 0.10 | 40.70±0.4 | 40.97±0.9 | 42.94±0.3 | 42.80±0.1 | 43.01±0.5 | - | 37.93±0.3 | 43.75±0.3 | **47.15**±0.4 | 46.38±0.4 | 41.85±3.1 | 43.75±0.3 | 46.66±0.1 | 46.75±0.1 | 45.87±0.3 | 46.21±0.1 | 46.69±0.1 | - | | 47.4 |
| | 0.50 | 42.90±0.4 | 44.06±0.3 | 44.54±0.4 | 43.86±0.5 | 43.46±0.8 | - | 37.76±0.4 | 43.30±0.4 | 46.71±0.2 | 46.38±0.2 | 33.39±6.0 | 45.05±0.3 | 46.69±0.4 | **47.01**±0.2 | 45.89±0.3 | 46.77±0.1 | 46.39±0.2 | - | | |
| | 1.00 | 42.62±0.2 | 44.51±0.3 | 44.68±0.6 | 45.12±0.4 | 43.53±0.6 | - | 37.66±0.3 | 34.39±6.0 | 43.84±0.4 | 46.13±0.2 | **46.61**±0.1 | 31.12±4.2 | 45.88±0.1 | 46.58±0.4 | **46.99**±0.1 | 45.81±0.1 | 46.12±0.2 | 46.24±0.3 | - | |
| Reddit | 0.05 | 40.00±1.1 | 45.83±1.7 | 40.13±0.9 | 46.88±0.4 | 40.24±0.8 | - | 88.23±0.1 | OOM | 69.96±0.5 | **90.63**±0.2 | 90.18±0.2 | OOM | 87.28±0.2 | 86.56±0.2 | 85.39±0.2 | 86.56±0.4 | 87.62±0.1 | 87.37±0.2 | - | 94.4 |
| | 0.10 | 50.47±1.4 | 51.22±1.4 | 55.73±0.5 | 59.34±0.7 | 48.28±0.7 | - | 88.32±0.1 | OOM | 76.95±0.2 | **91.33**±0.1 | 89.84±0.3 | OOM | 89.96±0.1 | 88.25±0.3 | 89.82±0.1 | 88.32±0.2 | 88.15±0.1 | 88.73±0.3 | - | |
| | 0.20 | 55.31±1.8 | 61.56±0.2 | 58.39±2.3 | 73.46±0.5 | 56.81±1.7 | - | 88.33±0.1 | OOM | 81.52±0.6 | **91.03**±0.3 | 90.71±0.1 | OOM | 89.08±0.1 | 88.73±0.2 | 90.42±0.1 | 88.84±0.2 | 87.03±0.1 | 90.65±0.1 | - | |
| Yelp | 0.05 | 48.67±0.3 | 46.81±0.1 | 46.08±0.0 | 46.08±0.0 | 46.07±0.0 | - | **55.04**±0.1 | 51.52±1.6 | 49.24±0.1 | 52.80±0.2 | 46.20±0.1 | OOM | 50.75±0.4 | 52.44±0.4 | 52.30±0.1 | 51.10±0.3 | 52.94±0.2 | 52.02±0.2 | - | 58.2 |
| | 0.10 | 51.03±0.4 | 46.08±0.0 | 46.28±0.1 | 52.23±0.3 | 46.22±0.0 | - | 53.51±0.8 | 51.68±1.0 | 47.33±0.5 | 47.56±0.2 | 47.96±0.0 | OOM | 52.49±0.1 | 49.70±0.1 | 53.22±0.1 | **52.54**±0.1 | 50.97±0.8 | 54.13±0.2 | - | |
| | 0.20 | 46.08±0.0 | 46.08±0.0 | 49.31±0.1 | 47.49±0.1 | 46.85±0.2 | - | 54.42±0.3 | 52.63±1.1 | 48.63±0.4 | 49.48±0.7 | 46.70±0.1 | OOM | **55.89**±0.2 | 48.77±1.3 | 51.76±0.2 | 52.19±0.5 | 51.35±0.5 | 52.86±0.1 | - | |

Table 10: Experiment results under **hyperparameter searching**. The search space is shown in Table 11. The best results, excluding the whole graph training results, are marked in **bold**.

| Dataset | Reduc. rate (%) | Coreset & Sparsification | | Coarsen | | Condensation | | | | | | Whole |
| | | | | | | Structure-free | | | Structure-based | | | |
| | | Random | K-Center | Averaging | VNG | GEOM | SFGC | GCondX | GCond | DosCond | SGDD | |
| Citeseer | 0.36 | 37.67±2.45 | 45.11±2.19 | 69.97±0.36 | 64.37±1.29 | 68.90±0.64 | 66.96±1.47 | 68.29±1.30 | **73.63**±0.32 | 69.53±0.65 | 71.90±0.24 | 72.6 |
| | 0.90 | 47.13±1.32 | 55.09±1.14 | 69.97±0.36 | 69.37±0.62 | **73.20**±0.35 | 70.66±0.23 | 69.73±0.46 | 70.93±0.51 | 70.97±0.29 | 70.10±0.73 | |
| | 1.80 | 64.21±0.72 | 62.82±0.78 | 70.01±0.27 | 69.35±0.70 | **74.36**±0.30 | 72.37±0.41 | 69.19±0.47 | 70.69±0.47 | 72.73±0.35 | 70.11±0.93 | |
| Cora | 0.50 | 47.93±0.96 | 49.92±3.06 | 76.55±0.91 | 70.61±0.64 | 79.03±0.61 | 76.80±2.18 | 80.04±0.60 | **80.63**±0.48 | 80.43±0.72 | 81.58±0.97 | 81.81 |
| | 1.30 | 69.54±2.60 | 63.16±1.37 | 76.99±0.67 | 75.72±0.21 | **83.10**±0.41 | 80.03±0.61 | 79.22±0.27 | 81.01±0.50 | 81.19±0.34 | 81.24±0.79 | |
| | 2.60 | 71.70±1.92 | 72.02±1.21 | 76.41±1.47 | 77.19±0.52 | **83.50**±0.43 | 81.64±0.53 | 78.98±0.31 | 81.45±0.46 | 81.06±0.53 | 79.80±0.85 | |
| Arxiv | 0.05 | 50.29±1.33 | 49.20±0.35 | 59.59±0.38 | 55.36±0.45 | 64.27±0.12 | **65.07**±0.49 | 59.63±0.37 | 55.83±0.68 | 56.74±0.36 | 59.13±0.45 | 71.22 |
| | 0.25 | 59.26±0.45 | 58.05±0.44 | 59.94±0.32 | 61.27±0.19 | **68.75**±0.10 | 66.63±0.28 | 62.43±0.31 | 64.79±0.27 | 57.56±0.22 | 56.86±0.42 | |
| | 0.50 | 62.49±0.75 | 60.77±0.37 | 59.93±0.29 | 64.78±0.13 | **69.63**±0.16 | 67.43±0.29 | 60.17±0.54 | 64.83±0.24 | 61.26±0.45 | 61.15±0.20 | |
| Flickr | 0.10 | 43.07±0.56 | 42.68±0.68 | 44.48±0.64 | 46.14±0.30 | **47.14**±0.11 | 46.93±0.25 | 46.74±0.12 | 46.63±0.11 | 45.92±0.19 | 46.79±0.14 | 47.4 |
| | 0.50 | 44.86±0.16 | 44.30±0.38 | 44.35±0.79 | 43.23±0.40 | 47.01±0.17 | **47.22**±0.15 | 46.76±0.10 | 47.13±0.14 | 46.20±0.18 | 46.38±0.15 | |
| | 1.00 | 45.63±0.24 | 44.70±0.47 | 44.38±0.78 | 43.97±0.52 | 46.93±0.24 | **47.02**±0.09 | 46.63±0.16 | 46.74±0.15 | 46.55±0.14 | 46.54±0.08 | |
| Reddit | 0.05 | 40.32±1.20 | 43.52±2.04 | 88.65±0.15 | 71.34±0.34 | **91.42**±0.08 | 90.18±0.14 | 86.92±0.26 | 86.53±0.21 | 86.66±0.15 | 87.71±0.20 | 93.95 |
| | 0.10 | 56.37±2.05 | 48.97±2.72 | 88.66±0.15 | 84.62±0.23 | **91.57**±0.04 | 89.88±0.19 | 88.37±0.35 | 87.81±0.22 | 88.44±0.15 | 88.88±0.25 | |
| | 0.20 | 63.56±1.08 | 56.27±2.99 | 88.60±0.34 | 89.03±0.14 | **91.57**±0.09 | 90.79±0.09 | 88.99±0.28 | 89.80±0.13 | 88.96±0.23 | 90.66±0.09 | |
| *# Wins after tuning* | | 0 | 0 | 0 | 0 | 10 | 3 | 0 | 2 | 0 | 0 | |
| *# Wins before tuning* | | 0 | 1 | 0 | 0 | 10 | 0 | 0 | 2 | 1 | 1 | |

### A.6.1 Details description for Test accuracy vs. total time Figure

Figure 1 compares test accuracy (y-axis) and total time (x-axis) for various graph condensation methods applied to the Arxiv dataset. The methods are distinguished by different marker shapes and colors: blue stars represent structure-free methods, red circles represent structure-based methods, and green triangles represent distribution-based methods. The size of each marker indicates the reduction rate, with smaller markers representing a reduction rate of 0.05%, medium markers 0.25%, and larger markers 0.50%. Dashed lines connect markers corresponding to the same method across different reduction rates, illustrating the method's behavior under varying levels of graph condensation. To enhance clarity, the name of each method will be positioned near the marker for its respective curve, ensuring easy identification of methods and their corresponding performance trends.

### A.6.2 Further analysis of experimental results

- **Factors Affecting Performance in Arxiv and Reddit.** We assume that the imbalanced label distributions in these two datasets are the factors for the performance. Arxiv and Reddit datasets have a larger number of classes and exhibit significant class imbalance compared to others. Consistent with most GC works, our implementation ensures at least one instance per class, guaranteeing representation for each class. However, this approach can cause distribution shifts. In contrast,

datasets like Cora, Citeseer, and Pubmed have more balanced training sets, leading to more stable performance. This observation highlights the need for improved initialization methods in the GC field to effectively handle datasets with numerous and imbalanced classes.

- **Why Averaging Achieves the Best Performance on Yelp.** This performance difference can be attributed to the characteristics of the *Yelp* dataset, which is designed for anomaly detection and evaluated using the F1-macro score. Averaging methods rely only on the average representations of normal instances and anomalies, resulting in a simple decision boundary that aligns well with the dataset's requirements. In contrast, GC methods may struggle due to unbalanced class initialization, often leading to overfitted decision boundaries for anomalies.

- **High performance variance across datasets or methods.** Some methods show high variance. A key example is MSGC, whose performance on Citeseer drops sharply from 72.08% to 60.24% at the 0.36% reduction rate. We hypothesize this instability stems from its reliance on multiple graph initializations, which can lead to inconsistent results depending on the run. we believe it's a design mismatch. trajectory-matching objective appears overly sophisticated for a In other cases, like the poorer performance of SFGC and GEOM on the Yelp dataset, Their simpler binary classification task, making them difficult to tune effectively. This instability highlights a potential sensitivity in certain designs. In contrast, methods like GCond and GCDM demonstrate much more consistent performance across different datasets and reduction rates. We will add this detailed analysis to our experimental section.

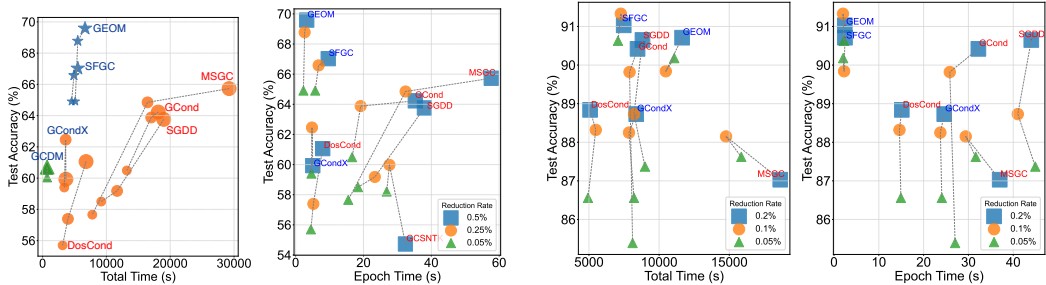

Figure 6: Performance vs. Total Time and Epoch Time on *Arxiv* (left two) and *Reddit* (right two).

## A.7 Privacy Preservation

### A.7.1 Step-by-step explanation of the MIA procedure

- **First**, an adversary trains a GCN model using only the publicly available condensed graph.
- **Next**, this adversary's model is used to generate an output confidence score for each node from the original graph (this includes both private training members and non-members).
- **Finally**, to measure the worst-case privacy leak, we perform a threshold analysis. We scan all possible confidence score thresholds to find the optimal value that maximizes the attack's accuracy in distinguishing members from non-members. This highest achievable accuracy is reported as the attack success rate.

### A.7.2 Discussions

We focused on a fundamental privacy attack, confidence-based membership inference attack (MIA), for the following reasons:

We are not merely benchmarking the privacy-preserving properties of existing GC methods but are also **broadening the scope of GC research** to encompass critical areas such as privacy and robustness. This expansion aims to demonstrate the potential of GC methods, inspiring **more researchers to recognize their promise and contribute to this emerging field**. Since existing applications of GC predominantly target Neural Architecture Search (NAS) [14, 25] and continual learning [70], we aim to shift the conversation by highlighting their broader applicability.

To the best of our knowledge, no prior work has empirically validated the privacy-preserving claims associated with GC. By targeting one of the most fundamental and well-studied privacy attacks, MIA, our work provides essential, empirical evidence for assessing and understanding the privacy

capabilities of GC. This serves as a **preliminary yet foundational step** toward establishing a systematic and rigorous framework for evaluating the privacy guarantees of GC methods. We have chosen to omit additional privacy attacks for the following reasons:

- **Model Inversion Attack (MIvA)** [71]: MIvA aims to reconstruct the original graph and assess attack performance via link prediction tasks. In the context of GC, the condensation process significantly reduces the number of nodes and reindexes all synthetic nodes. This reduction diminishes the granularity necessary for accurate link reconstruction, making it difficult for an attacker to determine specific node connections. Additionally, reindexing disrupts any direct correspondence between condensed and original nodes, further obfuscating the true link structure. Instead, we evaluate graph properties in Section 4.8, demonstrating that condensation alters most graph properties. This suggests that the privacy of graph properties is maintained through the condensed graph.
- **Attribute Inversion Attack (AIA)** [72]: AIA typically requires datasets with sensitive attributes, which diverges from the standard datasets in mainstream GNN studies [72, 73]. As a benchmark requiring unifying all baseline methods and datasets, Incorporating AIA would thus fall outside the scope of our current work.

## A.8   Transferability

### A.8.1   Hyperparameters Searching

For fair evaluation between different architectures, we conduct hyperparameter searching while training each architecture on the condensed graph. We select the best hyperparameter combinations based on validation results and report corresponding testing results. The search space of hyperparameters for each GNN is as follows: Number of hidden units is selected from {64, 256}, learning rate is chosen from {0.01, 0.001}, weight decay is 0 or 5e-4, dropout rate is 0 or 0.5. For GAT, since we fix the number of attention heads to 8, to avoid OOM, the number of hidden units is selected from {16, 64} and the search space of dropout rate is in {0.0, 0.5, 0.7}. Additionally, for SGC and APPNP, we also explore the number of linear layers in {1, 2}. For APPNP, we further search for alpha in {0.1, 0.2}.

### A.8.2   Hyperparameters Sensitivity Analysis

Figure 7: Hyperparameters Sensitivity Analysis on **Condensed Graphs**.

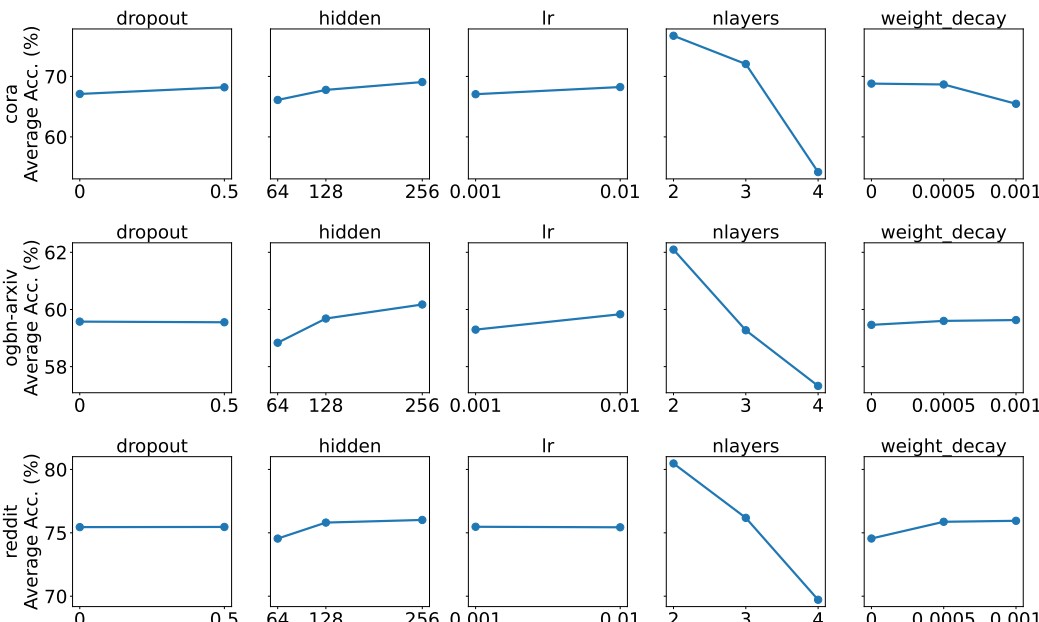

Figure 8: Hyperparameters Sensitivity Analysis on **Whole Graphs**.

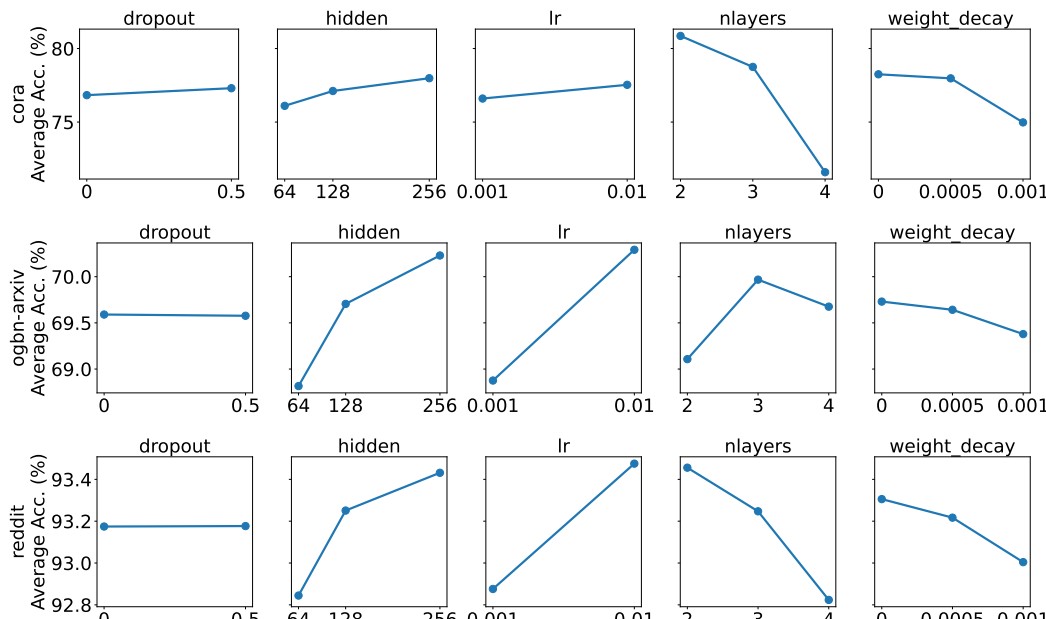

To provide additional insights on how varying hyperparameters affect the performance of the GNN model (e.g. GCN) trained on the whole or the condensed graphs, we further expand the search space of hyperparameters for GCN as shown in Table 11. The hyperparameter searching results for each method are shown in Table 10. We compare the winning times differences before and after tuning, which shows that GC methods that perform better in the main table generally maintain superior performance after hyperparameter tuning. Notably, methods like GEOM and GCond continue to outperform others post-tuning, reinforcing the robustness of our initial fixed hyperparameter choices.

Table 11: Hyperparameter Search Space for Sensitivity Analysis

| Hyperparameter | Values |
|---|---|
| Number of hidden units | {64, 128, 256} |
| Learning rate | {0.01, 0.001} |
| Number of layers | {2, 3, 4} |
| Weight decay | {0, 0.0005, 0.001} |
| Dropout rate | {0, 0.5} |

Figure 7 and 8 These figures show that condensed and whole graphs exhibit similar sensitivity patterns across the Cora, Ogbn-arxiv, and Reddit datasets, suggesting a consistent response to hyperparameter tuning.

- Both condensed and whole graphs show low sensitivity to **dropout and weight decay**, with minimal variations in accuracy, indicating these hyperparameters have a limited impact on performance.
- The **hidden layer size** positively influences accuracy in both condensed and whole graphs, with larger sizes generally improving performance, highlighting the importance of hidden layer capacity in model effectiveness.
- **Learning rate** sensitivity is also comparable between condensed and whole graphs; a higher learning rate (0.01) tends to perform better in both cases, though with slight dataset-specific variation (i.e. whole graph of Ogbn-arxiv).
- Notably, the **number of layers** impacts both graph types similarly, as accuracy consistently declines with an increase in layers, suggesting that deeper architectures do not benefit either condensed or whole graphs in three datasets.

Table 12: Property preservation check for GDEM, a method explicitly preserve the graph property.

| Dataset | Density % | Max Eigenvalue | DBI AGG | Homophily |
|---------|-----------|----------------|---------|-----------|
| *Cora* | 14.82 | 1.57 | 1.09 | 0.33 |
| Whole | 0.14 | 169.01 | 4.67 | 0.81 |
| *Citeseer* | 11.86 | 1.51 | 1.46 | 0.33 |
| Whole | 0.08 | 100.04 | 8.49 | 0.74 |
| *Pubmed* | 6.90 | 0.02 | 1.36 | 1.00 |
| Whole | 0.02 | 172.16 | 5.01 | 0.80 |

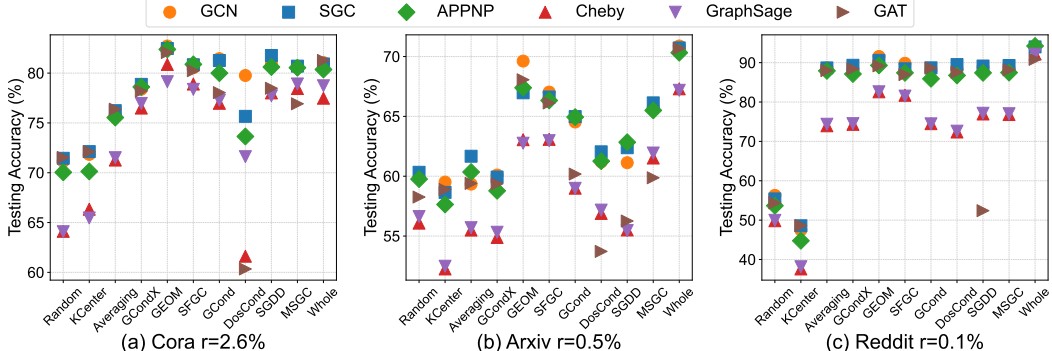

Figure 9: Performance of condensed graphs evaluated by different GNNs.

Thus, condensed and whole graphs have parallel sensitivity trends, where optimizing hidden layer size and learning rate while managing network depth is likely to enhance performance across both representations.

### A.8.3 Relative and Absolute Accuracy

We calculate the relative accuracy by dividing the results of the model trained on the condensed graph by the results of the same model trained on the whole graph. For example, the accuracy of GCN on the GCond condensed graph is divided by the accuracy of results on the whole graph. Since Figure 4 in the main content shows the relative accuracy, we show the absolute results of each GNN here in Figure 9.

### A.8.4 Evaluate Condensed Graph by Graph Transformer

The architectures discussed in the main content primarily utilize message-passing styles, which facilitate their transfer to each other. However, they may encounter challenges when applied to an entirely different architecture. Therefore, to conduct a more comprehensive evaluation of transferability, we assess the performance of various condensation methods using a graph transformer-based architecture **SGFormer** [56], which is totally different from those message-passing architectures. Figure 10 shows that SGFormer achieves comparable performance with other architectures on three non-GNN methods (Random, KCenter, Averaging). However, its performance significantly drops when trained on graphs condensed by GNN-involved methods. This suggests that future research should explore the transferability of other graph learning architectures.

### A.9 Neural Architecture Search

We utilize APPNP [54] for NAS experiments because its architecture modules are flexible and can be easily modified. The detailed architecture search space is shown in Table 14. Following the settings in GCond [14], we search full combinations of these choices, i.e., 480 in total for each dataset.

In addition to measuring the correlation between performance on the real graph and the condensed graph, we further analyze the preservation of top-performing architectures, which is a common

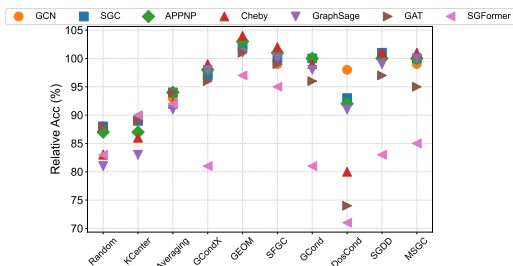

Figure 10: Condensed graph performance evaluated using different models including **SGFormer** on *Cora*.

practice in NAS studies. Specifically, we evaluate whether the ground-truth top-$K$ architectures on the real graph remain within the top-$K$ when evaluated on the condensed graph, quantified by Recall@$K$. We compute this metric by first ranking all candidate architectures on the real graph by performance and selecting the top $K$ as the ground-truth top-$K$ set, and then ranking the same candidates on the condensed graph to determine how many of the ground-truth top-$K$ architectures also appear in the top $K$ on the condensed graph.

Table 13: Recall@$K$ of top-$K$ architectures evaluated on Cora with $r = 0.5$.

|  | **Random** | **K-Center** | **GCondX** | **SFGC** | **GEOM** | **GCond** | **DosCond** | **MSGC** |
|---|---|---|---|---|---|---|---|---|
| Recall@5 | 0.0 | 0.0 | 0.2 | 0.0 | 0.2 | 0.0 | 0.0 | 0.0 |
| Recall@10 | 0.0 | 0.0 | 0.2 | 0.1 | 0.2 | 0.2 | 0.1 | 0.1 |
| Recall@20 | 0.4 | 0.35 | 0.5 | 0.45 | 0.4 | 0.45 | 0.25 | 0.4 |

We conduct experiments on Cora with $r = 0.5$. From Table 13, we observe that condensation methods incorporating gradient matching terms (GCondX, GCond) outperform coreset-based methods (Random, K-Center) in preserving the ground-truth top-$K$ architectures, suggesting that including performance gradient information better aligns relative performance between the real and condensed graphs. SFGC performs slightly below the gradient-based methods at moderate $K$ values (10 and 20), indicating that adding local structural consistency constraints can also improve fidelity. In contrast, purely random or center-based methods lack performance cues and achieve the lowest recall. Although recall values for all methods increase with $K$, the highest value reaches only 0.5 at $K = 20$, suggesting that condensed graphs still struggle to retain all high-performing architectures. Future work should further explore strategies that jointly optimize performance signals and graph structure to enhance consistency between condensed and real graphs.

### A.10  Graph property preservation

The full results on graph property preservation are listed in Table 15. As we mention in the main content, different GC methods show totally different behavior w.r.t. property preservation. **First**, VNG and SGDD tend to produce almost complete graphs linking each node pair. That also leads to a lower homophily, as they create more proportion of inter-class connections. **Second**, VNG performs best in property preservation, however, it shows suboptimal accuracy in Table 9. This suggests that the selected graph properties are unnecessary to maintain or to preserve as much as possible. **Third**, as the only method that creates sparse graphs, MSGC is unique among these methods except in the Homophily. From this point of view, we hold that homophily is very important for future research on *structure-based* GC since all structure-based methods behave consistently. Current research mostly holds the view that the loss of homophily is harmful [74], but our benchmark may provide a contradictory perspective on this.

Notably, we observed that MSGC preserves the maximum eigenvalue up to 0.94. As further evidence, the latest method, GDEM [37], focuses on learning to preserve eigenvectors, supporting the idea that maintaining spectral properties may be beneficial. However, upon closer examination of the properties of the graph synthesized by GDEM, as shown in Table 12, we find that these properties are not fully preserved. This is because their method only retains eigenvalues within a middle range,

Table 14: Architecture search space for APPNP.

| Architecture | Search Space |
|---|---|
| Number of propagation $K$ | {2, 4, 6, 8, 10} |
| Residual coefficient $\alpha$ | {0.1, 0.2} |
| Hidden dimension | {16, 32, 64, 128, 256, 512} |
| Activation function | {Sigmoid, Tanh, ReLU, Linear, Softplus, LeakyReLU, ReLU6, ELU} |

Table 15: Graph properties in condensed graphs from different *structure-based* GC methods. The "**Corr.**" row shows the correlation of certain properties between the condensed graph and the whole graph across five datasets.

| Graph property | Dataset and $r$ | VNG | GCond | MSGC | SGDD | Avg. | Whole |
|---|---|---|---|---|---|---|---|
| Density% | *Citeseer 1.8%* | 36.95 | 84.58 | 22.50 | 100.00 | 61.01 | 0.08 |
| (Structure) | *Cora 2.6%* | 52.17 | 82.28 | 22.00 | 100.00 | 64.11 | 0.14 |
| | *Arxiv 0.5%* | 100.00 | 75.40 | 8.17 | 99.91 | 70.87 | 0.01 |
| | *Flickr 1%* | 100.00 | 100.00 | 3.44 | 99.96 | 75.85 | 0.01 |
| | *Reddit 0.1%* | 100.00 | 2.67 | 32.07 | 74.85 | 52.39 | 0.05 |
| | **Corr.** | -0.81 | 0.07 | 0.55 | 0.13 | -0.01 | - |
| Max Eigenvalue | *Citeseer 1.8%* | 2.98 | 22.53 | 1.67 | 10.29 | 9.37 | 100.04 |
| (Spectra) | *Cora 2.6%* | 3.73 | 34.90 | 1.69 | 14.09 | 13.60 | 169.01 |
| | *Arxiv 0.5%* | 2,092.99 | 163.95 | 2.33 | 79.95 | 584.81 | 13,161.87 |
| | *Flickr 1%* | 1,133.94 | 281.04 | 1.76 | 123.86 | 385.15 | 930.01 |
| | *Reddit 0.1%* | 1,120.64 | 152.00 | 2.00 | 99.84 | 343.62 | 2,503.07 |
| | **Corr.** | 0.85 | 0.25 | 0.95 | 0.28 | 0.58 | - |
| DBI | *Citeseer 1.8%* | 4.14 | 1.40 | 1.98 | 3.47 | 2.75 | 12.07 |
| (Label & Feature) | *Cora 2.6%* | 3.69 | 1.84 | 0.70 | 4.34 | 2.64 | 9.28 |
| | *Arxiv 0.5%* | 2.27 | 2.62 | 2.49 | 2.80 | 2.55 | 7.12 |
| | *Flickr 1%* | 5.60 | 7.14 | 7.33 | 13.57 | 8.41 | 31.02 |
| | *Reddit 0.1%* | 1.51 | 2.16 | 1.49 | 1.53 | 1.67 | 9.59 |
| | **Corr.** | 0.81 | 0.93 | 0.94 | 0.97 | 0.91 | - |
| DBI-AGG | *Citeseer 1.8%* | 4.11 | 0.76 | 1.75 | 0.00 | 1.66 | 8.49 |
| (Label & Feature & Structure) | *Cora 2.6%* | 3.59 | 0.38 | 0.57 | 0.18 | 1.18 | 4.67 |
| | *Arxiv 0.5%* | 2.38 | 2.86 | 2.61 | 1.77 | 2.41 | 4.40 |
| | *Flickr 1%* | 20.26 | 11.60 | 7.90 | 6.51 | 11.57 | 25.61 |
| | *Reddit 0.1%* | 1.56 | 1.90 | 1.49 | 1.37 | 1.58 | 2.48 |
| | **Corr.** | 0.99 | 0.93 | 0.95 | 0.89 | **0.94** | - |
| Homophily | *Citeseer 1.8%* | 0.18 | 0.18 | 0.23 | 0.15 | 0.18 | 0.74 |
| (Label & Structure) | *Cora 2.6%* | 0.14 | 0.16 | 0.19 | 0.13 | 0.16 | 0.81 |
| | *Arxiv 0.5%* | 0.08 | 0.07 | 0.04 | 0.07 | 0.07 | 0.65 |
| | *Flickr 1%* | 0.34 | 0.27 | 0.27 | 0.27 | 0.29 | 0.33 |
| | *Reddit 0.1%* | 0.04 | 0.04 | 0.04 | 0.07 | 0.05 | 0.78 |
| | **Corr.** | -0.83 | -0.68 | -0.46 | -0.80 | -0.69 | - |

specifically from $K_1$ to $K_2$. This suggests that methods for accurately preserving spectral properties remain an area for further exploration.

Since only the metric DBI does not rely on structure, we also exhibit the correlation of DBI of structure-free methods across all five datasets in Table 16. From the comparison between structure-free and structure-based methods, we find that GCondX and GEOM also preserve this correlation of DBI to some extent, similar to structure-based methods.

**Obs. 12: Preserving Density%, DBI, and Homophily tends to be beneficial for downstream tasks.** Table 5 indeed measures property preservation, defined as the Pearson correlation between property values in the condensed graphs and the original graphs. This measure does not indicate the relationship between properties and downstream accuracy. To clarify this connection, we have conducted an additional analysis, shown in Table 17. In this table, the Correlation column reports the Pearson correlation between the average test accuracy across five datasets (row "Avg ACC") and

Table 16: DBI in condensed graphs from both *structure-based* and *structure-free* GC methods, continued from Table 15.

| Datasets | VNG | GCond | MSGC | SGDD | GCondX | GEOM | Avg. | Whole |
|---|---|---|---|---|---|---|---|---|
| *Citeseer 1.8%* | 4.14 | 1.40 | 1.98 | 3.47 | 2.90 | 2.55 | 2.74 | 12.07 |
| *Cora 2.6%* | 3.69 | 1.84 | 0.70 | 4.34 | 2.18 | 3.16 | 2.65 | 9.28 |
| *Arxiv 0.5%* | 2.27 | 2.62 | 2.49 | 2.80 | 5.52 | 4.37 | 3.35 | 7.12 |
| *Flickr 1%* | 5.60 | 7.14 | 7.33 | 13.57 | 22.93 | 6.04 | 10.43 | 31.02 |
| *Reddit 0.1%* | 1.51 | 2.16 | 1.49 | 1.53 | 0.57 | 2.96 | 1.70 | 9.59 |
| **Corr.** | 0.81 | 0.93 | 0.94 | 0.97 | 0.95 | 0.78 | 0.90 | - |

each property preservation metric (Density%, Max Eigen, DBI, DBI-AGG, and Homophily). These results indicate that Density%, DBI, and Homophily show stronger associations with downstream performance, whereas Max Eigen and DBI-AGG do not.

Table 17: Correlation between condensed graph properties and model performance.

| | VNG | GCond | MSGC | SGDD | Correlation |
|---|---|---|---|---|---|
| Avg ACC | 63.34 | 69.40 | 69.02 | 68.95 | - |
| Density (%) | -0.81 | 0.07 | 0.55 | 0.13 | 0.91 |
| Max Eigen | 0.85 | 0.25 | 0.95 | 0.28 | -0.51 |
| DBI | 0.81 | 0.93 | 0.94 | 0.97 | 0.96 |
| DBI-AGG | 0.99 | 0.93 | 0.95 | 0.89 | -0.80 |
| Homophily | -0.83 | -0.68 | -0.46 | -0.80 | 0.54 |

## A.11 Denoising effects

All corruptions are implemented by a library for attack and defense methods on graphs, DeepRobust [75]. The full results on denoising effects are in Table 18. Apart from GC methods, we also add coreset selection methods as baselines. Results show that the simple baseline, Random, contains a certain level of denoising effects in terms of performance drop in *Citeseer* and *Flickr*. Meanwhile, KCenter exhibits the lowest performance drop in *Cora* corrupted by structural noise and adversarial structural attack. However, these phenomena do not necessarily mean they can defend the attack as the performance of these two methods before being corrupted is worse than GC methods. In contrast, the GC methods naturally outperform whole graph training in most scenarios, even though they are not specifically designed for defense.

## A.12 Code Availablity and Usage

We have developed an easy-to-use code package, which is included in the supplementary material and has been open-sourced as a PyTorch library. The package accepts graphs in the PyG (PyTorch Geometric) format as input and outputs a reduced graph that preserves the properties or performance of the original graph. Below, we provide technical details on how users can integrate new datasets, implement their own methods, propose new settings, and address potential difficulties.

### A.12.1 Usage

```
from graphslim.dataset import *
from graphslim.evaluation import *
from graphslim.condensation import GCond
from graphslim.config import cli

args = cli(standalone_mode=False)
# Customize arguments here
args.reduction_rate = 0.5
args.device = 'cuda:0'
```

Table 18: Denoising effects of selected methods. "Perf. Drop" shows the relative loss of accuracy compared to the original results of each method before being corrupted. The best results are in **bold** and results that outperform whole dataset training are underlined. *Structure-free* and *structure-based* methods are colored as blue and red.

| Dataset | Method | Feature Noise | | Structural Noise | | Adversarial Structural Noise | |
|---|---|---|---|---|---|---|---|
| | | Test Acc. ↑ | Perf. Drop ↓ | Test Acc. ↑ | Perf. Drop ↓ | Test Acc. ↑ | Perf. Drop ↓ |
| *Citeseer 1.8%* (Poisoning & Evasion) | Whole | 64.07 | 11.75% | 57.63 | 20.62% | 53.90 | 25.76% |
| | Random | 56.91 | 9.11% | 61.56 | **1.69%** | 59.42 | 5.12% |
| | KCenter | 52.80 | 10.57% | 55.41 | 6.15% | 55.07 | 6.73% |
| | GCond | **64.06** | **7.63%** | **65.64** | 5.35% | **66.19** | **4.55%** |
| | GCondX | 61.27 | 10.40% | 60.42 | 11.65% | 60.75 | 11.15% |
| | GEOM | 58.77 | 19.53% | 51.41 | 29.60% | 57.94 | 20.67% |
| *Cora 2.6%* (Poisoning & Evasion) | Whole | 74.77 | 8.26% | 72.13 | 11.49% | 66.63 | 18.24% |
| | Random | 59.89 | 17.10% | 62.64 | 13.28% | 65.33 | 9.57% |
| | KCenter | 59.88 | 15.13% | 62.94 | **10.79%** | 65.51 | **7.14%** |
| | GCond | 67.62 | 16.04% | 63.14 | 21.61% | 68.90 | 14.45% |
| | GCondX | **67.72** | **13.85%** | **63.95** | 18.63% | 69.24 | 11.91% |
| | GEOM | 49.68 | 40.01% | 53.59 | 35.29% | 66.32 | 19.93% |
| *Flickr 1%* (Poisoning) | Whole | 46.68 | 1.51% | 42.60 | 10.13% | 44.44 | 6.24% |
| | Random | 44.33 | **0.78%** | 43.28 | 3.13% | 43.93 | **1.69%** |
| | KCenter | 43.15 | 0.88% | 42.36 | 2.68% | 42.21 | 3.03% |
| | GCond | **46.29** | 1.49% | **46.97** | **0.04%** | 43.90 | 6.58% |
| | GCondX | 45.60 | 2.11% | 46.19 | 0.83% | 42.00 | 9.83% |
| | GEOM | 45.38 | 1.63% | 45.52 | 1.32% | **44.72** | 3.06% |

```
10  # Add more args.<main_args/dataset_args> as needed
11
12  graph = get_dataset('cora', args=args)
13  # To reproduce the benchmark, use our args and graph class
14  # To use your own args and graph format, ensure the args and graph
        class have the required attributes
15
16  # Create an agent for the reduction algorithm
17  # Add more args.<agent_args> as needed
18  agent = GCond(setting='trans', data=graph, args=args)
19
20  # Reduce the graph
21  reduced_graph = agent.reduce(graph, verbose=True)
22
23  # Create an evaluator
24  # Add more args.<evaluator_args> as needed
25  evaluator = Evaluator(args)
26
27  # Evaluate the reduced graph on a GNN model
28  res_mean, res_std = evaluator.evaluate(reduced_graph, model_type='GCN'
        )
```

Listing 1: Code Example for Using the Benchmark Package

### A.12.2 Customization

**Adding a New Dataset:** To implement a new dataset, create a new class in `dataset/loader.py` and inherit from the `TransAndInd` class.

**Implementing a New Reduction Algorithm:** To add a new reduction algorithm, create a new class in `sparsification`, `coarsening`, or `condensation`, and inherit from the `Base` class.

**Adding a New Evaluation Metric:** To implement a new evaluation metric, create a new function in `evaluation/eval_agent.py`.

**Implementing a New GNN Model:** To add a new GNN model, create a new class in `models` and inherit from the `Base` class.

### A.12.3 Potential Difficulties

Users may encounter the following challenges:

**Disk Space Limitations:**

- Some methods store training trajectories of multiple experts, which can exceed 100 GB.
- *Solution:* Reduce the number of experts using the `<method>.reduce()` module to manage disk space.

**Memory and GPU Constraints:**

- Larger datasets might cause memory or GPU limitations during the condensation process.
- *Solution:* Load data and adjust the reduction process to run in a mini-batch manner to reduce memory usage.

**Hyperparameter Adjustment:**

- Tuning hyperparameters may be necessary for optimal performance.
- *Solution:* Modify the JSON configuration files in the `configs` folder, which contain all hyperparameters for each method.

We believe this information will help users effectively utilize, customize, and integrate our benchmark package with new datasets or algorithms. We provide comprehensive documentation and support for easy adoption and extension.

### A.13 Practical guidelines for applying GC

In response to the request for practical guidance on applying GC in constrained or privacy-sensitive settings, our conclusions are directly supported by the observed trade-offs (Obs) and experiments results in the paper:

- **Condenser choice:** From Obs. 2 (Sec. 5.2) and Fig. 4, structure-free methods (GCondX, GCDM) are much more efficient in time and memory while maintaining competitive accuracy, which makes them better for low-resource devices.

- **Trajectory matching:** Obs. 1 (Sec. 5.1) and Fig. 3 show that trajectory-matching methods (GEOM, SFGC) yield the highest accuracy but require expensive preprocessing. A practical workflow is to condense graphs on a powerful machine and deploy the result on smaller systems.

- **Condensation ratio:** Sec. 6.1 and Fig. 6 indicate that GC remains effective even with 1 sample per class, but aggressive ratios may lead to memory errors. Ratios between 0.5 % and 2 % are stable.

- **Privacy:** Obs. 7 (Sec. 7.2) and Table 2 confirm that GC reduces membership inference attack success by 5–10 % with little accuracy loss, providing a simple privacy-preserving preprocessing step.

- **Noisy graphs:** Obs. 5 (Sec. 6.3) and Fig. 9 show GC improves robustness to structural noise, with structure-based condensers giving stronger gains. GC has limited benefit for feature noise, so other denoising methods are needed in those cases.

- **Transferability:** Obs. 8 (Sec. 8) shows gradient- and trajectory-matching improve neural architecture search and transfer across backbones, while structure-free methods are more scalable when hardware is limited.

### A.14 Benefits to Graph Machine Learning Community

Our benchmark and its insights offer significant benefits to the broader graph machine learning community in the following areas:

**(a) Current Position of GC in Graph Machine Learning.** First, GC originated in the computer vision domain but has been adapted to address the unique challenges of graph data. It incorporates

techniques from graph sampling and coarsening to effectively manage the complexities inherent to graph modalities while to extract essential information. Second, from the view of representation learning, GC aims to create a compact representation of the original graph, preserving essential features for training well-generalized GNNs. Third, GC is gaining traction due to its advantages in accelerating training, enhancing scalability, and improving visualization, making it a valuable tool for various graph-based applications such as NAS [25], continual learning [70] and explainability [76].

**(b) Addressing Key Questions.**

- **When and Why Specific GC Methods Work:** Our benchmark systematically evaluates different GC methods, elucidating the conditions under which each method excels. This helps researchers and users understand the strengths and limitations of various condensation techniques.
- **Broader Applications of GC:** We demonstrate the versatility of GC beyond traditional applications like NAS and continual learning. Our benchmark highlights its potential in areas such as privacy preservation and efficient data management.
- **Key Observations and Novel Insights:** Based on our well-established benchmark, we have made several new observations and provided fresh insights in the field of GC. For instance, GC methods exhibit significant denoising capabilities against structural noise but are less effective at mitigating node feature noise. Additionally, trajectory matching and gradient-based inner optimization are crucial for achieving reliable performance in NAS and enhancing transferability. These findings highlight both the strengths and limitations of current GC techniques.

**(c) Facilitating General Graph Machine Learning Research.**

- Our benchmark provides a pioneering investigation into the practical effectiveness of GC methods in privacy preservation and their denoising effects (robustness). This highlights the potential of GC methods to serve as a novel set of baselines for comparison with existing privacy defense and robustness techniques. Furthermore, as graph condensation inherently involves modifying datasets, i.e., a data-centric approach, it can be seamlessly integrated with model-centric efforts to deliver complementary benefits in robustness and privacy preservation.
- **Observation 4:** Certain GC methods can achieve both privacy preservation and high condensation performance. This dual capability suggests the potential to break the traditional trade-off between privacy and utility in the trustworthy graph learning area by effectively synthesizing data.
- **Observation 7:** We observe that different GC methods exhibit varying degrees of transferability across datasets, indicating natural differences among GNNs including Graph Transformer. This inspires a rethinking of the similarities between current GNN models, particularly regarding the perspectives and priors they prefer to extract.
- **Observation 11:** We observed that homophilous graphs often become heterophilous after condensation while still maintaining high performance. This unexpected outcome challenges the conventional understanding of the relationship between GNN performance and homophily [77]. Our findings suggest that the dependency of GNNs on homophily may need to be reevaluated, opening new avenues for research into how graph condensation affects structural properties and model performance.

Overall, our benchmark serves as a valuable resource for graph machine learning researchers by providing comprehensive evaluations, uncovering new applications of GC, and inspiring innovative methodologies. This facilitates advancements in the field, enabling the creation of more effective and adaptable graph learning models.

