# OpenReview forum: "GC4NC: A Benchmark Framework for Graph Condensation on Node Classification with New Insights"
_NeurIPS.cc/2025/Datasets_and_Benchmarks_Track — NeurIPS 2025 Datasets and Benchmarks Track poster_

### Official Review · Reviewer_qLeK · 2025-06-24

**Rating:** 5
**Confidence:** 4

**Summary:**

In this work, the authors present a comprehensive analysis of various graph condensation methods.

To this end, they evaluate 15 condensation (and coarsening) techniques across 7 datasets using a range of evaluation metrics. Their analysis examines these algorithms from multiple perspectives, including neural architecture search (NAS), privacy preservation, and transferability.

Through this thorough investigation, the authors uncover several noteworthy insights into the behavior of graph condensation algorithms.

**Additional Feedback:**

N/A

**Dataset Code Accessibility:**

Yes

**Dataset Code Comments:**

The code and datasets are well presented in GitHub.

**Ethical Considerations:**

No, there are no or only very minor ethics concerns

**Final Justification:**

I think this work has sufficient contribution to be accepted in NeurIPS.

**Limitations Weaknesses:**

**[NAS evaluation metric]** While assessing the correlation between performance on the real graph and the condensed graph is an important evaluation metric, I believe it would be more practical to focus on analyzing the top-performing architectures, as is common in many NAS studies. Specifically, evaluating whether the ground-truth top-K architectures remain in the top-K on the condensed graph—measured by recall—would provide more actionable insights.

**[Privacy metric]** To me, it is unclear why identifying training nodes in the condensed graph serves as a metric for privacy risk. Providing further justification for this choice would help readers better understand its relevance to the work.

**[Table and figure arrangement]** I find that the current version of the manuscript is quite dense and not optimally organized. For example, the analysis results and their corresponding tables are placed far apart. While I understand this may be due to space constraints, I believe a better arrangement of the content would make it easier for readers to follow the analyses and results. I hope this will be improved in the camera-ready version.

**Strengths Contributions:**

**[Comprehensive analysis]** The authors offer a thorough analysis of condensation methods, addressing areas that were previously unexplored.

**[Clear comparison with existing benchmarks]** In the Appendix, they present a systematic and comprehensive comparison with existing benchmark graph condensation works.

**[Clean implementation]** The implementation code is well-organized and user-friendly.

---

> ### Author Rebuttal · Authors · 2025-07-31
>
> Thanks a lot for your valuable comments.
> > **Q1: While assessing the correlation between performance on the real graph and the condensed graph is an important evaluation metric, I believe it would be more practical to focus on analyzing the top-performing architectures, as is common in many NAS studies. Specifically, evaluating whether the ground-truth top-K architectures remain in the top-K on the condensed graph—measured by recall—would provide more actionable insights.**
> >
>
> |  | **Random** | **K-Center** | **GCondX** | **SFGC** | **GEOM** | **GCond** | **DosCond** | **MSGC** |
> | --- | --- | --- | --- | --- | --- | --- | --- | --- |
> | Recall@5 | 0 | 0 | 0.2 | 0 | 0.2 | 0 | 0 | 0 |
> | Recall@10 | 0 | 0 | 0.2 | 0.1 | 0.2 | 0.2 | 0.1 | 0.1 |
> | Recall@20 | 0.4 | 0.35 | 0.5 | 0.45 | 0.4 | 0.45 | 0.25 | 0.4 |
>
> Thank you to the reviewer for the suggestion. **We agree that focusing on top‑K architectures has practical value.**
>
> We compute **Recall@K** by first ranking all candidate architectures on the real graph by performance and selecting the top K as the ground‑truth top‑K set. We then rank the same candidates on the condensed graph and count how many of the ground‑truth top‑K architectures also appear in the top K of the condensed graph.
>
> We conduct experiments on Cora with r=0.5. From **above table** we have the following observations. **First**, condensation methods with gradient matching terms (GCondX, GCond) outperform coreset methods (Random, K‑Center) in preserving the ground‑truth top‑K architectures, which indicates that introducing performance gradient information helps align relative performance between the real graph and the condensed graph. **Second**, SFGC performs just behind the gradient methods at moderate K values (10, 20), showing that adding local consistency constraints in structure aggregation also improves fidelity. **Third**, pure random or center‑based methods lack performance signals and remain at the lowest recall levels. **Finally**, although recall for all methods increases with K, the best method achieves only 0.5 at K=20, indicating that the condensed graph still fails to retain all high‑performing architectures and that future work should further explore joint optimization of performance signals and graph structure.
>
> We will include these experimental results and the detailed analysis in **Appendix A.8**.
>
> > **Q2: To me, it is unclear why identifying training nodes in the condensed graph serves as a metric for privacy risk. Providing further justification for this choice would help readers better understand its relevance to the work.**
> >
>
> We measure privacy risk via a membership‑inference attack (MIA) on the condensed graph S: an adversary who sees S tries to guess, for each node in the original graph, whether it was used during condensation (i.e. in the training set).
>
> This directly captures the risk that S encodes training‑specific information. In the machine‑learning privacy literature, MIA accuracy is the standard way to quantify how much an attacker can infer about membership of data points (e.g. [45] uses confidence scores on labels to distinguish train vs. test samples). By applying the same idea to nodes, **lower MIA accuracy means S leaks less about which original nodes were used—hence better privacy preservation**.
>
> We will add this explanation at the end of **Section 3.1** (and refer explicitly to **Appendix A.6** where we justify the choice of MIA over other attacks), to make the connection between “identifying training nodes” and “membership inference” crystal clear.
>
> > **Q3: I find that the current version of the manuscript is quite dense and not optimally organized. For example, the analysis results and their corresponding tables are placed far apart. While I understand this may be due to space constraints, I believe a better arrangement of the content would make it easier for readers to follow the analyses and results. I hope this will be improved in the camera-ready version.**
> >
>
> Thank you for pointing this out. We agree that the current layout is rather dense and that some analysis results are separated from their tables by several pages. In the next version, we will use the additional page allowance to reorganize the content more clearly. Specifically, we will:
>
> 1. Move **Table 2** and **Table 3** so that each appears directly above its corresponding **Section 4.2** and **Section 4.3**, respectively.
> 2. **Increase line spacing** as well as the spacing between figures and surrounding text and between figures and their captions.
> 3. Relocate **Figure 4** to appear immediately before **Section 4.6**.
> 4. Place **Figure 5** adjacent to **Section 4.5** for easier cross‑reference.
>
> These changes will improve the readability and flow of our analysis and results presentation. We hope this can help address your concern.

---

> > ### Comment · Reviewer_qLeK · 2025-08-01
> >
> > Thank you for addressing my questions.
> > I think this work has a sufficient contribution, and therefore, I would like to maintain my score (accept) from the positive side.

---

### Official Review · Reviewer_Wz8v · 2025-06-28

**Rating:** 4
**Confidence:** 3

**Summary:**

The paper proposes GC4NC, an evaluation framework for graph condensation (GC) methods on node classification tasks. It offers a fair evaluation protocol, an open-source codebase, and a multi-dimensional comparison across performance, efficiency, privacy, denoising, NAS, and transferability. The authors also perform in-depth analysis of design choices (e.g., initialization, structure usage).

**Additional Feedback:**

1. Have the authors tested whether results hold across different GNNs?
2. Is the framework modular enough to easily plug in new backbones?
3. Any heterophilic datasets (e.g., Chameleon, Squirrel, Texas) included? If not, can the authors comment on how representative their dataset selection is?

**Dataset Code Accessibility:**

Yes

**Ethical Considerations:**

No, there are no or only very minor ethics concerns

**Final Justification:**

My main concerns have been addressed.

**Limitations Weaknesses:**

1. Single model backbone (GCN) may bias the benchmark. Since many GC methods are designed or tuned for specific backbones (e.g., GNTK), using only GCN for evaluation may introduce unfair bias.
2. Limited graph diversity. If all selected datasets are homophilic (e.g., Cora, Citeseer), the benchmark misses important real-world cases where heterophily dominates.

**Strengths Contributions:**

1. GC is an active field. This work fills a clear gap by providing the first unified benchmark.
2. Covers not just accuracy but also privacy preservation, denoising, transferability, and NAS effectiveness which often been overlooked in prior work.
3. The paper gives non-trivial observations (e.g., trajectory matching, structure-free trade-offs) that can inform the community.

---

> ### Author Rebuttal · Authors · 2025-07-31
>
> Many thanks to Reviewer Wz8v for providing the detailed comments.
> > **Q1: Single model backbone (GCN) may bias the benchmark. Since many GC methods are designed or tuned for specific backbones (e.g., GNTK), using only GCN for evaluation may introduce unfair bias. Have the authors tested whether results hold across different GNNs?**
> >
>
> We have pointed out your concern about the model backbone bias in **(e) Transferability** in “**Multi-Dimensional Evaluation**” in **Section 3.1**. The experiments that test whether results hold across different GNNs are conducted in **Section 4.5.**
>
> From **Figure 5** we can observe that there is **no significant performance loss** for the **majority** of cases when condensed graphs are transferred to various GNNs. This highlights the success of GC methods, which typically **only use GCN or SGC for condensation**.
>
> > **Q2: Limited graph diversity. If all selected datasets are homophilic (e.g., Cora, Citeseer), the benchmark misses important real-world cases where heterophily dominates. Any heterophilic datasets (e.g., Chameleon, Squirrel, Texas) included? If not, can the authors comment on how representative their dataset selection is?**
> >
>
> Thank you for this crucial suggestion. Evaluating performance on heterophilic graphs is very important, and we appreciate the opportunity to clarify our experimental choices.
>
> We aimed to cover this by including two large-scale benchmarks that exhibit lower homophily:
>
> - **Flickr**, which is distinctly heterophilic with a homophily ratio of **0.33**.
> - **ogbn-arxiv**, which can be considered semi-heterophilic at a **0.65** homophily ratio.
>
> We did consider other well-known heterophilic datasets like the ones you mentioned (Chameleon, Squirrel, Texas). However, we made a principled decision not to include them. The core motivation of graph condensation is to make training on **large-scale** graphs more efficient. Applying these methods to **small graphs** does not effectively test the scalability and performance benefits that our benchmark aims to measure.
>
> Your point is extremely well-taken. The behavior of condensation methods in low-homophily regimes is a critical and underexplored area. We will explicitly add a discussion to our paper acknowledging this as a limitation and highlighting the need for more comprehensive heterophilic benchmarks as a vital direction for future work.
>
> > **Q3: Is the framework modular enough to easily plug in new backbones?**
> >
>
> Yes. Along with the benchmark, we provide a PyTorch library for graph reduction named **GraphSlim**, which is explicitly designed to be modular and backbone-agnostic.
>
> **Backbone integration.** In GraphSlim, every backbone is defined inside `graphslim/models` and inherits from a common `BaseGNN` class. Adding a new backbone requires only two steps:
>
> 1. Create a new Python file that subclasses `BaseGNN`.
> 2. Add one import line in `graphslim/models/__init__.py`.
>
> **No changes are needed elsewhere.** The framework **automatically** detects and uses the new model. The current implementation already integrates diverse backbones such as GCN, GraphSage, GAT, APPNP, and SGFormer, showing that this plug-in mechanism works reliably.
>
> **Method integration (condensation, coarsening, sparsification).** GraphSlim organizes reduction algorithms into three pipelines: `sparsification`, `coarsening`, and `condensation`. Adding a new method **only requires creating a Python file** that subclasses the corresponding base class (such as `GCondBase`) and registering it with **one import line** in that directory’s `__init__.py`. The framework then loads it **automatically** when the method name is passed through the command line (`--method`). **No other changes are needed.** This mechanism is already validated by more than **30 integrated algorithms**.
>
> GraphSlim is both backbone-agnostic and method-agnostic. Extending any pipeline, whether by adding a new backbone or a new graph reduction method, requires **minimal effort** and **does not require changes** to the core framework.

---

> > ### Comment · Reviewer_Wz8v · 2025-08-05
> >
> > Thank you for the detailed response. My main concerns have been addressed.

---

> > > ### Author Response · Authors · 2025-08-06
> > >
> > > Thank you for your feedback. We are glad to hear that your major concerns have been addressed! It would be great if you could consider reflecting this by increasing the score. We are also happy to clarify further concerns.

---

> ### Author Response · Authors · 2025-08-04
>
> As the discussion phase is approaching itsend, we kindly request the reviewer to let us know if the above clarifications have addressed the remaining questions. We would be happy to address any additional points the reviewer may have during the remaining time of the discussion phase. We thank the reviewer for engaging in the discussion with us.

---

### Official Review · Reviewer_xJYU · 2025-06-30

**Rating:** 4
**Confidence:** 3

**Summary:**

This paper introduces a comprehensive benchmark for graph condensation, it analyzed the impact of various methods, modules, and evaluation metrics across the entire graph condensation pipeline. The authors provide detailed empirical comparisons and extract insights from a variety of angles, including model performance, privacy preservation, denoising ability, and hardware constraints.

**Dataset Code Accessibility:**

Yes

**Ethical Considerations:**

No, there are no or only very minor ethics concerns

**Final Justification:**

I believe the authors have addressed my main concerns.

**Limitations Weaknesses:**

Major:

While the authors aim to summarize key insights from the experimental results, the phrasing is relying heavily on vague terms such as “certain/some/...” While this sounds safe and avoids overgeneralization, it also weakens the clarity and informativeness of the conclusions.

The paper appears to assume the reader is highly familiar with the field of graph condensation. However, the goal of a benchmark is typically to serve a broader community by facilitating easier comparison and reproducibility of new models, methods, or tasks. So I recommend that the authors at least include a brief introduction to the overall workflow of graph condensation training and evaluation, and clarify key terminology when it first appears (e.g., GNTK). Also, the organization of the paper can be difficult to follow due to undefined terms. For example, it is unclear what “image condensation” is, why this notion is introduced, and how it fits into the comparison. Similarly, terms like “random” and “herding” seem to serve as both coreset selection and initialization methods, but this is not clearly explained.

When evaluating denoising capabilities, the authors first inject artificial noise into the data and then try to denoise it. This is a little bit confused to me. Does this imply that the original dataset is noise-free? Or is it because the noise in the original data cannot be measured or quantified? If the latter, on what basis can you assume and verify that noise exists in the original dataset? In this context, denoising appears to be a "created" capability. Therefore, the authors need to justify both the rationale and necessity of evaluating denoising before drawing meaningful conclusions from this experiment.

When evaluating the privacy preservation capability, what is the specific procedure used? Why is it necessary to determine whether nodes from the training set appear in the condensed graph? more specific: why do training nodes appear in the condensed graph?

Minor:

Table 1 and Figure 1 suggest that structure-free methods are both more efficient and more accurate. What then is the advantage or motivation for using structure-based methods?

In Figures 3 and 5, overlapping data points obscure the specific performance values for each method.

Given the many evaluation dimensions compared in the benchmark, it would be helpful to explicitly describe how other variables are held constant in each comparison.

**Strengths Contributions:**

The paper studies the entire graph condensation pipeline, including reduction ratio, initialization strategy, and graph property preservation. It also considers both algorithmic performance and practical concerns such as computation cost and hardware adaptability.

Beyond accuracy, the authors explore important "soft power" aspects like privacy preservation, denoising capability, and suitability for neural architecture search, it's a well-rounded benchmark.

---

> ### Author Rebuttal · Authors · 2025-07-31
>
> Many thanks to Reviewer xJYU for providing the detailed comments.
> >Q1: While the authors aim to summarize key insights from the experimental results, the phrasing is relying heavily on vague terms such as “certain/some/...” While this sounds safe and avoids overgeneralization, it also weakens the clarity and informativeness of the conclusions.
>
> We thank the reviewer for this excellent suggestion. We agree that replacing vague terms with specific findings in the headers of our observations will make our contributions more impactful.
>
> We would like to clarify that our use of this phrasing in the observation headers was a deliberate choice. The headers are intended to provide a high-level summary, while the specific, detailed evidence supporting these claims is provided immediately following in the main text. This structure was chosen to ensure scientific precision, as our benchmark revealed highly nuanced results where behaviors were not universal across all methods or datasets. Stating a highly specific claim in the header would often be too narrow, while a universal claim would be an inaccurate overgeneralization.
>
> However, we fully agree with the reviewer that the headers themselves can and should be made more informative. As suggested, we will revise our observation statements to be more direct, serving as a better bridge to the detailed explanations.
>
> - *Before: “Obs.4: Certain GC methods can achieve both privacy preservation and high condensation performance.”*
> - **After:** *“Obs.4: Certain GC Methods such as TM and eigen-decomposition -based GC methods can achieve a strong balance between privacy preservation and condensation performance.”*
>
> This change will be applied throughout the paper to strengthen our claims.
>
> >Q2: The paper appears to assume the reader is highly familiar with the field of graph condensation. However, the goal of a benchmark is typically to serve a broader community by facilitating easier comparison and reproducibility of new models, methods, or tasks. So I recommend that the authors at least include a brief introduction to the overall workflow of graph condensation training and evaluation, and clarify key terminology when it first appears (e.g., GNTK). Also, the organization of the paper can be difficult to follow due to undefined terms. For example, it is unclear what “image condensation” is, why this notion is introduced, and how it fits into the comparison. Similarly, terms like “random” and “herding” seem to serve as both coreset selection and initialization methods, but this is not clearly explained.
>
> We sincerely thank the reviewer for this valuable feedback on improving the paper's accessibility. To better serve a broader audience:
>
> - We'll add a new background section to introduce the standard **graph condensation workflow**. We'll also define key terms like **GNTK, image condensation** when they first appear, so readers aren't left guessing.
> - We'll clarify the **image condensation** in the Section 1. We will cite **DC-bench [16]** earlier and explain in detail in Section 4.1: highlighting the performance gap where image methods on CIFAR10 at **IPC=1** get ~**50%** accuracy compared to the whole dataset training, while most GC methods on Cora can exceed **98%**.
> - We'll fix the confusion about **random** and **herding**. As we note in **Section 2.2**, they are coreset methods. We have mentioned in **Section 3.2** (**Line 198**) that all corset methods can also be used for initialization. We will clearly mention these coreset methods here.
>
> >Q3: When evaluating denoising capabilities, the authors first inject artificial noise into the data and then try to denoise it. This is a little bit confused to me. Does this imply that the original dataset is noise-free? Or is it because the noise in the original data cannot be measured or quantified? If the latter, on what basis can you assume and verify that noise exists in the original dataset? In this context, denoising appears to be a "created" capability. Therefore, the authors need to justify both the rationale and necessity of evaluating denoising before drawing meaningful conclusions from this experiment.
>
> Our approach does not assume the original data is noise-free. Rather, we employ a standard methodology from the broader machine learning and graph learning fields[1] to evaluate **robustness**. By intentionally injecting a known quantity and type of noise (e.g., structural or adversarial), we create a controlled setting. This is the one way to systematically and fairly measure how different condensation methods withstand specific perturbations, making our comparisons rigorous and reproducible.
>
> [1] Jin, Wei, et al. "Graph structure learning for robust graph neural networks." *Proceedings of the 26th ACM SIGKDD international conference on knowledge discovery & data mining*. 2020.
>
> >Q4: When evaluating the privacy preservation capability, what is the specific procedure used? Why is it necessary to determine whether nodes from the training set appear in the condensed graph? more specific: why do training nodes appear in the condensed graph?
>
> We appreciate the opportunity to clarify our privacy evaluation.
>
> We have a procedure in **Section 4.2** illustrating that we use confidence scores and scan the threshold. We will expand it with a more detailed, step-by-step explanation of the procedure.
>
> - **First**, an adversary trains a GCN model using only the publicly available condensed graph.
> - **Next**, this adversary's model is used to generate an output **confidence score** for each node from the original graph (this includes both private training members and non-members).
> - **Finally**, to measure the worst-case privacy leak, we perform a **threshold analysis**. We scan all possible confidence score thresholds to find the optimal value that maximizes the attack's accuracy in distinguishing members from non-members. This highest achievable accuracy is reported as the attack success rate.
>
> For the second question, we have to clarify a crucial point: **the original training nodes do not appear directly in the condensed graph.** The condensed graph is composed of newly synthesized nodes whose features and structure are optimized to distill the full training set. The privacy risk emerges because this distillation process might "imprint" recoverable information about the private training data onto the publicly shared condensed graph.
>
> >Q5: Table 1 and Figure 1 suggest that structure-free methods are both more efficient and more accurate. What then is the advantage or motivation for using structure-based methods?
>
> This is an excellent and insightful question that highlights a key goal of our benchmark and we have a detailed analysis in **Section 5 Conclusion and Outlook**.
>
> Our results do indeed establish that for the primary metrics of **performance and efficiency**, structure-free methods currently lead the field.
>
> However, the motivation for considering structure-based methods lies in their performance on other critical dimensions. Our benchmark reveals that certain structure-based methods offer superior trade-offs for specific use cases, such as:
>
> - **Better robustness** to noise (denoising).
> - Stronger performance in **Neural Architecture Search (NAS)** tasks.
> - **Privacy concern**.
>
> Our goal is to provide a multi-faceted view that helps practitioners select a method based on their complete set of priorities. We will sharpen this point in the paper's conclusion.
>
> >Q6: In Figures 3 and 5, overlapping data points obscure the specific performance values for each method.
>
> We promise to revise Figures 3 and 5 in the next version. We will use techniques like point jittering and distinct marker styles to ensure all data points are clearly visible and interpretable.
>
> >Q7: Given the many evaluation dimensions compared in the benchmark, it would be helpful to explicitly describe how other variables are held constant in each comparison.
>
> You have raised a crucial point about experimental control. We have used a unified pipeline (Section 4.1) where each GC method is paired with its author-recommend hyperparameters including the initialization. The resulting condensed graph is then evaluated with a **standardized GCN architecture and standard GCN hyperparameters**, ensuring a fair comparison of the condensed graphs themselves.
>
> - **Privacy and NAS:** We have mentioned we use the condensed graph in Line 156 and Line 178. For most subsequent analyses, we use the **fixed condensed graphs** from the main evaluation. We then vary only the specific evaluation protocol (e.g., the membership inference classifier for privacy).
> - **Initialization** **and Denoising**. For the former, we fix the other parts of condensation methods following the Section 4.1 and vary only the initialization. For the latter, we fix the entire pipeline but vary the input data by adding noise.
>
> We will make this explicit in the revised manuscript.
>
> We want to sincerely thank you again for your detailed and constructive review. Your insightful feedback has provided us with a clear roadmap for significantly improving our manuscript. We are confident that these revisions will address all your concerns and make our paper stronger, clearer, and more impactful for the community. We appreciate your time and consideration.

---

> > ### Author Response · Authors · 2025-08-04
> >
> > We would like to thank you again for your valuable suggestions. As the discussion phase draws to a close, we wish to confirm whether our clarifications have sufficiently addressed your concerns. If so, we would appreciate your consideration in updating your score to reflect these improvements. We are happy to clarify any question you may have.

---

> > ### Comment · Reviewer_xJYU · 2025-08-04
> >
> > Thank you for your response. I believe my main concerns have been addressed, and I have updated my score.

---

### Official Review · Reviewer_gVA4 · 2025-07-03

**Rating:** 5
**Confidence:** 3

**Summary:**

This paper presents GC4NC, a benchmark framework for evaluating graph condensation (GC) methods, with a focus on node classification. The framework addresses several key limitations in the current GC literature, including inconsistent evaluation protocols, limited benchmarking dimensions, and underexplored design choices. GC4NC evaluates GC methods across multiple axes: performance, efficiency, privacy preservation, denoising ability, neural architecture search (NAS) effectiveness, and transferability. The authors provide thorough empirical results, a unified evaluation protocol, and an open-source implementation.

**Additional Feedback:**

1. Consider generalizing GC4NC to support broader GNN tasks beyond node classification.

2. Provide more detailed discussion on failure cases or unexpected trends (e.g., why structure-free methods underperform under noise).

3. Offer practical guidelines for applying GC in resource-constrained or privacy-sensitive environments, based on the observed trade-offs.

**Dataset Code Accessibility:**

Yes

**Dataset Code Comments:**

The code and datasets can be available at their provided link.

**Ethical Considerations:**

No, there are no or only very minor ethics concerns

**Final Justification:**

All of my concerns has been addressed.

**Limitations Weaknesses:**

1. The benchmark is currently limited to node classification; extending it to other graph tasks (e.g., link prediction, graph classification) would improve its scope.

2. Some results show high performance variance across datasets or methods (e.g., with structure-free vs. structure-based GC), which may need further clarification.

3. As mentioned in the paper, it currently does not incorporate privacy-preserving techniques like differential privacy, despite the emphasis on privacy evaluation. Current privacy results are limited to membership inference attacks.

4. Graph property preservation is evaluated for a few structural and spectral properties, but the connection between property preservation and downstream performance is not yet deeply explored.

**Strengths Contributions:**

1. The paper addresses a pressing need in the GC literature by establishing a standardized evaluation protocol for benchmarking GC methods on NC tasks.

2. Evaluates methods across multiple underexplored axes (e.g., privacy, denoising, NAS), providing practical insights into GC applicability.

3. The framework is well-documented and reproducible, with open-source code available.

4. The authors conduct extensive empirical comparisons and provide analysis of design choices (e.g., structure-free vs. structure-based, initialization strategies).

---

> ### Author Rebuttal · Authors · 2025-07-31
>
> Many thanks to Reviewer gVA4 for providing thorough, insightful comments.
>
> > **Q1: The benchmark is currently limited to node classification.**
> >
>
> We agree that extending the benchmark to a wider range of graph-related tasks is a **valuable direction for future research**. However, the decision to focus GC4NC exclusively on **node classification (NC)** was a deliberate design choice intended to maximize the benchmark's **immediate impact and analytical depth** for the following reasons:
>
> 1. **Alignment with the Current State of the Field**: We chose to focus on node classification because the vast majority (**approximately 90%**) of existing graph condensation papers have concentrated on this specific task [17,13,29,30,31,34,39,38]. As a benchmark, our goal is to serve the primary needs of the community by providing a standardized and rigorous evaluation framework for the most common application of this technique.
> 2. **Enabling Depth Over Breadth**: This focused approach allows for a deeper and more comprehensive analysis than a broader, multi-task benchmark would permit. This depth enabled us to pioneer evaluations along several critical dimensions that have been previously under-explored: **(a)** Our framework is the **first** to empirically test the inherent denoising abilities of GC methods against feature, structural, and adversarial noise; **(b)** We are the **first** to assess the privacy-preserving capabilities of various GC methods under MIA attack.
> 3. **Recognized Scope and Future Directions**: Our choice of scope was intentional. We note in our appendix that "we do not include graph classification method such as KiDD [59]" because our focus is NC. We also explicitly acknowledge this as a potential area for expansion in our conclusion, stating that future work includes "examining the transferability of condensed knowledge to various domains and downstream tasks".
>
> > **Q2: High performance variance across datasets or methods.**
> >
>
> That's a great observation. Analyzing this variance is crucial for understanding the real-world utility between different GC methods.
>
> Some methods show **high variance**. A key example is **MSGC**, whose performance on Citeseer drops sharply from **72.08% to 60.24%** at the 0.36% reduction rate. We hypothesize this instability stems from its reliance on multiple graph initializations, which can lead to inconsistent results depending on the run. we believe it's a **design mismatch**.  trajectory-matching objective appears overly sophisticated for a In other cases, like the poorer performance of **SFGC** and **GEOM** on the Yelp dataset, Their complexsimpler binary classification task, making them difficult to tune effectively. This instability highlights a potential sensitivity in certain designs. In contrast, methods like **GCond** and **GCDM** demonstrate much more **consistent performance** across different datasets and reduction rates. We will add this detailed analysis to our experimental section.
>
> > **Q3: Not incorporate privacy-preserving techniques like differential privacy.**
> >
>
> We focus on two representative attacks for privacy evaluation in our benchmark (we explain why in **Appendix A.6**): the **fundamental privacy attack**, which provides a basic assessment of privacy leakage, and the **confidence-based membership inference attack (MIA)**, a widely adopted method that uses model confidence to infer whether a node was part of the training set. These attacks capture the most direct and commonly studied form of privacy risk in graph learning, and thus serve as a starting point for systematic comparison.
>
> **We appreciate the suggestion to** **explore additional privacy-preserving techniques such as DP**. To provide a reference, we implement a simple DP baseline by adding Gaussian noise to the outputs of test nodes (standard deviation = 0.1 for Cora and Citeseer, and 1.0 for Arxiv). The results are shown below:
>
> | Methods | Cora MIA Acc (↓) | Cora Acc (↑) | Citeseer MIA Acc (↓) | Citeseer Acc (↑) | Arxiv MIA Acc (↓) | Arxiv Acc (↑) |
> | --- | --- | --- | --- | --- | --- | --- |
> | **Whole** | 74.87 ± 1.16 | 81.50 ± 0.50 | 81.76 ± 1.01 | 72.61 ± 0.27 | 54.26 ± 0.11 | 71.43 ± 0.11 |
> | **DP** | *75.49 ± 0.91* | *80.60 ± 0.56* | *83.82 ± 1.13* | *67.17 ± 1.03* | *54.35 ± 0.18* | *64.39 ± 0.22* |
> | **GCond** | 72.10 ± 0.96 | 80.54 ± 0.67 | 74.11 ± 0.61 | 69.35 ± 0.82 | **53.04 ± 0.18** | 64.23 ± 0.16 |
> | **GDEM** | **60.66 ± 1.26** | 81.76 ± 0.53 | 70.01 ± 2.94 | 71.74 ± 0.90 | - | - |
> | **GEOM** | 67.90 ± 0.55 | **82.82 ± 0.17** | **67.55 ± 0.62** | **73.03 ± 0.31** | 53.80 ± 0.19 | **69.59 ± 0.24** |
>
> The results show that most GC methods offer a **superior privacy-utility trade-off** compared to baseline Differential Privacy (DP).
>
> On **Cora** and **Citeseer**, GC methods generally provide both higher test accuracy and stronger privacy (a lower MIA success rate). The exception is on **Arxiv**, where the performance trade-offs between GC and DP are much more comparable.
>
> This confirms that GC approaches can reduce privacy risks without explicit DP mechanisms and remain a competitive alternative to noise-based methods.
>
> > **Q4: Connection between property preservation and downstream performance is not yet deeply explored.**
> >
>
> Thank you for pointing this out, as it helps us improve the paper.
>
> **Table 5** indeed measures ***property preservation***, defined as the Pearson correlation between property values in the condensed graphs and the original graphs. This measure does **not** indicate the relationship between **properties and downstream accuracy**. To clarify this connection, we have conducted an additional analysis, shown below:
>
> |  | **VNG** | **GCond** | **MSGC** | **SGDD** |  |
> | --- | --- | --- | --- | --- | --- |
> | **Avg ACC** | 63.34 | 69.40 | 69.02 | 68.95 | **Correlation** |
> | **Density%** | -0.81 | 0.07 | 0.55 | 0.13 | **0.91** |
> | ***Max Eigen*** | 0.85 | 0.25 | 0.95 | 0.28 | **-0.51** |
> | **DBI** | 0.81 | 0.93 | 0.94 | 0.97 | **0.96** |
> | **DBI-AGG** | 0.99 | 0.93 | 0.95 | 0.89 | **-0.80** |
> | **Homophily** | -0.83 | -0.68 | -0.46 | -0.80 | **0.54** |
>
> In this table, the **Correlation** column reports the Pearson correlation between the **average test accuracy across five datasets** (row “Avg ACC”) and each **property preservation metric** (Density %, Max Eigen, DBI, DBI‑AGG, and Homophily). These results indicate that Density %, DBI, and Homophily show stronger associations with downstream performance, whereas Max Eigen and DBI‑AGG do not.
>
> We will update **Table 5** in the revised manuscript to include this correlation column so that the relationship between property preservation and downstream performance is made explicit. In addition, **Section 4.7** will be extended with a new observation (**Obs. 12**) stating that **preserving Density %, DBI, and Homophily tends to be beneficial for downstream tasks**.
>
> > **Q5: Provide discussion on failure cases or unexpected trends.**
> >
>
> Our experiments show that structure-free methods degrade more under noise as shown in **Table 3** and discussed in **Obs. 5** in **Section 4.3**. The main reason is that they do not learn a synthetic adjacency matrix and thus cannot benefit from **structural smoothing** to mitigate incorrect edges. When random or adversarial edges are added to the original graph, these methods cannot use message passing to reduce the effect of such perturbations. In contrast, structure-based methods learn both node features and a synthetic structure, and the **learned structure** itself acts as a **denoising mechanism**, which makes them more robust to structural perturbations. For feature noise, both types of methods drop significantly because the current objectives have no explicit mechanism to handle corrupted features.
>
> > **Q6: Practical guidelines for applying GC in resource-constrained or privacy-sensitive environments, based on the observed trade-offs.**
> >
>
> In response to the request for practical guidance on applying GC in constrained or privacy-sensitive settings, our conclusions are directly supported by the **observed trade-offs** (Obs) and **experiments results** in the paper:
>
> - **Condenser choice:** From **Obs. 2 (Sec. 5.2)** and **Fig. 4**, **structure-free** methods (GCondX, GCDM) are much more efficient in time and memory while maintaining competitive accuracy, which makes them better for low-resource devices.
> - **Trajectory matching:** **Obs. 1 (Sec. 5.1)** and **Fig. 3** show that **trajectory-matching** methods (GEOM, SFGC) yield the highest accuracy but require expensive preprocessing. A practical workflow is to condense graphs on a powerful machine and deploy the result on smaller systems.
> - **Condensation ratio:** **Sec. 6.1 and Fig. 6** indicate that GC remains effective even with 1 sample per class, but aggressive ratios may lead to memory errors. Ratios between **0.5 % and 2 %** are stable.
> - **Privacy:** **Obs. 7 (Sec. 7.2)** and **Table 2** confirm that GC reduces **membership inference attack** success by 5–10 % with little accuracy loss, providing a simple privacy-preserving preprocessing step.
> - **Noisy graphs:** **Obs. 5 (Sec. 6.3)** and **Fig. 9** show GC improves robustness to **structural noise**, with **structure-based** condensers giving stronger gains. GC has limited benefit for feature noise, so other denoising methods are needed in those cases.
> - **Transferability:** **Obs. 8 (Sec. 8)** shows **gradient- and trajectory-matching** improve neural architecture search and transfer across backbones, while **structure-free** methods are more scalable when hardware is limited.
>
> We promise we will include this discussion in a new section named “**Practical guidelines for applying GC in resource-constrained or privacy-sensitive environments**” in our paper’s appendix.

---

> > ### Comment · Reviewer_gVA4 · 2025-08-03
> >
> > Thank you for your detailed response. All of my concerns have been addressed, so I have updated my score from 4 (Weak Accept) to 5 (Accept).

---

### Note · Authors · 2025-08-12

Dear AC,

Thank you so much for your great efforts. For your convenience, we would like to summarize the reviews and highlight our main contributions below.

**Reviews:** All four reviews are now positive.

- Before rebuttal: **gVA4** scored 4, **Wz8v** 4, **qLeK** 5, and **xJYU** 3.
- After rebuttal:
    - **gVA4** said “All of my concerns have been addressed, I have updated my score from 4 (Weak Accept) to **5 (Accept)**”.
    - **Wz8v** said “My main concerns have been addressed”.
    - **qLeK** said “I think this work has a sufficient contribution, and therefore, I would like to maintain my score **5 (accept)**”.
    - **xJYU** said “I believe my main concerns have been addressed, and **I have updated my score**”.
- As a result, all reviewers now express a positive stance toward acceptance.

**Main Contributions:**

- **Unified benchmark** for graph condensation on node classification with a fair and consistent evaluation protocol.
- **Multi-dimensional evaluation** covering performance, efficiency, privacy, denoising, NAS, transferability, and key design choices (structure-free vs. structure-based, initialization, property preservation).
- **Novel Insights.** Through a comprehensive comparison of these methods, our experimental results provide key insights into the behavior of graph condensation
- **Open-source and extensible framework** (GraphSlim) supporting new backbones and methods with minimal integration cost.

In light of the now fully positive reviews, we believe our work is a strong candidate for acceptance at NeurIPS and will make a valuable contribution to the graph machine learning community.

Best,

Authors

---

### Decision · Program_Chairs · 2025-09-18

**Decision:**

Accept (poster)

**Comment:**

This paper introduces a unified benchmark for evaluating graph condensation methods in node classification. The authors address inconsistent protocols and limited evaluation scope by assessing fifteen methods across seven datasets with metrics such as efficiency, privacy, NAS effectiveness, and transferability. Their thorough empirical study provides clear insights into the strengths and limitations of current approaches. After the rebuttal, reviewers’ concerns were fully addressed.